# Combining a Standardized Growth Class Assessment, UAV Sensor Data, GIS Processing, and Machine Learning Classification to Derive a Correlation with the Vigour and Canopy Volume of Grapevines

**DOI:** 10.3390/s25020431

**Published:** 2025-01-13

**Authors:** Ronald P. Dillner, Maria A. Wimmer, Matthias Porten, Thomas Udelhoven, Rebecca Retzlaff

**Affiliations:** 1Department of Viticulture and Oenology, DLR (Dienstleistungszentrum Ländlicher Raum) Mosel/Steillagenzentrum, Gartenstraße 18, 54470 Bernkastel-Kues, Germany; matthias.porten@dlr.rlp.de; 2Department of Computer Science, University of Koblenz, Universitätsstraße 1, 56070 Koblenz, Germany; wimmer@uni-koblenz.de; 3Department of Environmental Remote Sensing and Geoinformatics, Trier University, Universitätsring 15, 54296 Trier, Germany; udelhove@uni-trier.de (T.U.); retzlaff@uni-trier.de (R.R.)

**Keywords:** precision viticulture, vine vigour, remote sensing, UAV (Unmanned Aerial Vehicle), photogrammetry, multispectral imagery, geodata processing, GIS (Geoinformation Systems), machine learning, ground truth

## Abstract

Assessing vines’ vigour is essential for vineyard management and automatization of viticulture machines, including shaking adjustments of berry harvesters during grape harvest or leaf pruning applications. To address these problems, based on a standardized growth class assessment, labeled ground truth data of precisely located grapevines were predicted with specifically selected Machine Learning (ML) classifiers (Random Forest Classifier (RFC), Support Vector Machines (SVM)), utilizing multispectral UAV (Unmanned Aerial Vehicle) sensor data. The input features for ML model training comprise spectral, structural, and texture feature types generated from multispectral orthomosaics (spectral features), Digital Terrain and Surface Models (DTM/DSM- structural features), and Gray-Level Co-occurrence Matrix (GLCM) calculations (texture features). The specific features were selected based on extensive literature research, including especially the fields of precision agri- and viticulture. To integrate only vine canopy-exclusive features into ML classifications, different feature types were extracted and spatially aggregated (zonal statistics), based on a combined pixel- and object-based image-segmentation-technique-created vine row mask around each single grapevine position. The extracted canopy features were progressively grouped into seven input feature groups for model training. Model overall performance metrics were optimized with grid search-based hyperparameter tuning and repeated-k-fold-cross-validation. Finally, ML-based growth class prediction results were extensively discussed and evaluated for overall (accuracy, f1-weighted) and growth class specific- classification metrics (accuracy, user- and producer accuracy).

## 1. Introduction

Detecting relevant vineyard objects, including vine rows, grapevine stems, and quantifying vine health properties, like vigour, are essential for effective vineyard management [1,2,3]. Therefore, extracting vines and correlating important viticultural parameters, like grape yield and vigour, with sensor-derived image features is essential for high-precision vineyard management and automatization approaches. For this purpose, various sensor technologies and methods from sensor-based precision viticulture are at the forefront [4]. These technologies include different sensor-system data processed with new digital techniques, including classical image processing to more recent machine learning and deep learning techniques [5,6,7]. According to the vineyard-specific geometric structure and size, high-resolution UAV (Unmanned Aerial Vehicle)-derived images and proximal sensors are favored compared to aerial and satellite sensor data in viticulture applications [8,9]. Based on these sensor systems, many approaches have been developed over recent decades for extracting vineyard objects, especially vine rows and single vines. For vine rows, these approaches include manual GIS (Geographic Information System)-based extraction methods in combination with DTM/DSM (Digital Terrain Model/Digital Surface Model) data [10], threshold-based GIS- methods [11,12], object-based image segmentation techniques [13,14,15,16], HSV (Hue Saturation Value)-based methods [17], texture-based techniques [18], geometric structure extraction from stereo image techniques [19], ultrasonic sensor detection techniques [20], LiDAR (Light Detection And Ranging)- point cloud processing techniques [21,22,23] deep learning methods [24], and combinations of various methods [15,17,25]. Moreover, several vine extraction approaches were developed based on data from different sensor types and processing routines [26,27,28,29,30,31,32].

The extracted objects comprised different crops [33,34], including vines [35], subsequently joined and correlated to the extracted image features from the sensor data. For example, some authors derived canopy parameters like the LAI (Leaf Area Index) [36], health status of vines [37], or the canopy volume [27] from different sensor types and data sources. In the case of 2D multispectral image data, these features comprise essentially spectral features like Vegetation Indices (VI) [38], structural features like CHM (Canopy Height Model), or texture features [18]. The extracted features are correlated to the ground truth data [39,40,41,42,43], which were measured and labeled in the field to generate the training and evaluation datasets as basis for model training and evaluation. The selection of the prediction technique depends on the specific data structure and type of prediction variable. Therefore, several methods are used, from classical regression methods [43] to more sophisticated modern machine learning and deep learning-based approaches, depending on those constraints [44,45].

Under consideration of recent research in sensor-based precision agri- and viticulture, the research objective of this study is to evaluate the potential of combining canopy spectral, structural, and texture vine canopy features for growth estimation of vines based on machine learning model predictions of specific vine growth classes, applying the growth class assessment of Porten [43] as ground truth. Hitherto, no previous study used this specific assessment and UAV- image data geoprocessing methodology for machine learning classifier-based predictions of vine vigour classes.

To reach this objective, the following research questions and objectives of this study were formulated:Is there a significant difference between spectral, structural, and texture input features and their combinations for model prediction metrics and the robustness of the results?Which classifiers after hyperparameter tuning provide the best overall prediction performance?Which growth classes have the highest positive prediction matches, and which have the lowest matches? Is there a pattern in class-specific metrics that can be addressed in the model prediction process and the selected input feature groups?Based on the findings from the previous points, what are the advantages and limitations of the growth class assessment after Porten [43], and how could the evaluation, database, methodology, and classification process be optimized to improve the prediction and separability of the growth classes in the future?

To answer these research questions, the generation and extraction of different canopy feature types (spectral, structural, texture) of single vines based on UAV-multispectral sensor data, Digital Terrain and Surface Models (DTM/DSM), and derived geodata (e.g., CHM- Canopy Height Model) were investigated as prediction variables for growth classifications. To achieve this goal, the most suited variables, such as Vegetation Indices (VI) (spectral features), height measures (e.g., Canopy Height Model-structural features), and texture features (GLCM- Gray-Level Co-occurrence Matrix), were selected, computed and considered as model training input to predict the growth classes after [43], which were used as target classes. To fit the training data, the most promising machine learning classifiers were selected based on extensive literature research in the fields of precision agriculture, viticulture, and environmental sciences using sensor-based remote and proximal sensor techniques for crop, yield and biomass estimation. Moreover, selected ML classifier hyperparameters were selected and tuned with a grid-search-based repeated-k-fold-cross-validation approach. Next, the quality of class separation due to the input feature groups, classifier selection, and robustness of the prediction results was examined, based on chosen model metrics and the current literature on biomass, crop, and vine volume and vigour estimation. Thereafter, the class-specific accuracy (confusion matrix) was computed and evaluated for a more differentiated evaluation of class-specific influences, related to the input feature types and input feature groups. Finally, the growth classification assessment after Porten [43] and the methodology of this study will be briefly evaluated in terms of the potential for machine learning-based growth predictions of grapevines, including future perspectives, for enhancing the classification scores and improving the growth class separation procedure.

## 2. Materials and Methods

The following subsections outline the study’s investigation area, materials, and methods as the foundation for subsequent processing and classification routines.

### 2.1. Test Site and Field Data Acquisition (Ground Truth Data)

The investigation area is located in the so-called arena, a test vine field of the DLR (*Dienstleistungszentrum Ländlicher Raum*, a service center for rural area based in Bernkastel-Kues) Mosel, in the northwest of Bernkastel-Kues in Rhineland, Palatinate, (Germany) (see Figure 1) (49.91334° N, 7.04960° E (EPSG: 4326- WGS 84)). The approximately 2 ha sized area consists of regularly sloping vine rows in fall line (slope orientation) with some individual vine row tips.

The area has an average slope of approx. 30° and a predominantly SSW orientation at an average altitude between 183–208 m above sea level. The vine variety under investigation is White Riesling (*Vitis vinifera subsp. vinifera*), a typical grapevine variety of the Moselle region. The form of training is espalier training with two pendulum arches (see Figure 2).

The average distance between vine rows is approximately 2.0 m, and the average distance between adjacent vines in the same row is approximately 1.20 m (standing space 2.4 m^2^/vine). During ground truth labeling (see Section 2.4) and UAV-dataset generation (see Section 2.5.1 and Section 2.5.2) at véraison stage (03.08.2022), the mean temperature was recorded with 30.5 °C, with a relative humidity of 34% and a wind speed of around 1 m/s at the local weather station (see source link in Appendix A). Typical grapevine infections within the Moselle area are the vine leaf disease Bois Noir [46], the grapevine trunk disease Esca [47], and virus infections [48], which can affect vital parameters like vigour.

### 2.2. Vine Growth and Growth Classification Assessment After Porten (2020)

The vigour of grapevines is influenced by several key factors, such as soil composition, soil chemistry, soil management, vine variety, fertilization type and amount, stand space, rootstem, and gate level [49]. The concept of terroir and its relation to vine quality is more explicitly discussed by [50,51,52]. Therefore, vine vigour as an explicit expression of vine health depends on many interacting factors, including the vegetative and reproductive phenological growth stages. Moreover, vigour is a good indicator of vegetative and reproductive parameters like canopy volume, height, leaf area, grape yield, and related vineyard management and cultivation strategies [29,53,54]. Therefore, due to the potentially high correlation of vigour with physiological parameters, the growth class assessment after [43] was selected as basis for the ground truth target classes for later correlation and ML classification with the processed, extracted, and aggregated UAV-sensor image data. The specific growth classification assessment after Porten [43] has not been used before, for correlation and classification with UAV-sensor derived image data. Previously, less differentiated assessments were correlated with UAV-sensor data [55,56,57,58,59,60] on whole vine sites, but without considering the growth stages on the spatial resolution of each single grapevine stem. Nevertheless, for the effective correlation with UAV-sensor image data, a differentiated vine growth assessment is necessary to describe the internal structure of each vine described by [61]. Under consideration of these previous studies and requirements, the growth class assessment after [43] was developed, extending previously used assessments for correlation with UAV-sensor data. For validation, [43] tested the assessment on several vine fields by correlating the labeled vines with relevant viticulture vegetative and reproductive parameters including dry wood weight, one-year wood, leaf chlorophyll content, or the number of bunches measured on the same vines. The correlation and statistical significance of the growth class labels and the relevant parameters were investigated using several statistical methods, including factor analysis and statistical significance tests like one-way ANOVA (Analysis of Variance) and the Mann–Whitney-U-test. For comparison, similar correlations and spatial interpolations of viticulture parameters with different remote and proximal sensor types for investigating vineyard variability were achieved by [62]. Therefore, the growth class assessment after [43] is proven to be scientifically valid and can be used as a template for ground truth data labeling regarding remote- and proximal-sensor-based growth predictions. The specific structure and coding of the assessment are described in the following section.

### 2.3. Growth and Infection Classification Assessment After Porten (2020)

The “growth” in growth classes can be seen as a synonym for gradual vine vigour categories, from no growth (0) to excellent growth (9), with the intermediate growth classes (1,3,5,7) in between. For each growth class a color code was used, for visualizing the grapevine stem positions together with the growth classes categories, determined in the field (see first column in Table 1).

The gap between growth classes (3 to 5) does not necessarily express a linear relationship to vine vigour; rather, it provides space for intermediate classes if modifications are required to adapt or extend the assessment. For the visual standardization of the growth classes, template pictures of each growth class from a side view were taken (see Figure 3). This is especially important to reduce labeling error sources, like intra- and interpersonal errors, as much as possible. Within the classification process, only growth classes without infection type and infection strength were considered. Nevertheless, the infection has a strong impact on vine vigour [43] but was not the main focus of attention in this investigation. Accordingly, the specific characteristics of the individual infections are not described in detail here because this was not the focus of this study.

### 2.4. Manual Ground Truth Labeling of Vines

Based on the standardized classification scheme after [43], the vines were labeled during the véraison stage (3 Auguest 2022) and merged with the vines position described in the method section (see Section 2). The total number of vines labeled amounts to 1657, with a growth class count distribution of growth class 0 = 206, growth class 1 = 190, growth class 3 = 343, growth class 5 = 470, growth class 7 = 402, growth class 9 = 46.

For this study, only growth class labels without the infection additive and infection strength were considered as target classes for machine learning predictions due to the overall weak correlation and explanatory power of infection to vine growth concerning all vines. The véraison was chosen as reference time for labeling due to the most significant differentiation of viticultural parameters concerning the individual growth classes [43], especially during the onset of bunch’ ripening. Labeling of the vines was done simultaneously to UAV-sensor data acquisitions and terrestrial field data measurements (soil water content, early leaf water potential) for meaningful correlation. The growth class ground truth was used as target variable for the training-test data split previous to fitting the preprocessed sensor image data to the ML classifiers. In terms of UAV-sensor-based ML classification, ground truth data included vine yield measurements [8], leaf water potential [60], stem water potential [41], and stomatal conductance [42], not considering growth as the target variable, as in this study. Within this study, the early morning leaf water potential was measured before sunset at 2–3 am. Besides bivariate correlation the soil water content and leaf water measurement data were not further considered for ML classification of the growth classes due to only a few sampling sites and inappropriate measurement effort.

### 2.5. UAV Data Acquisition, Preparation, and Processing

#### 2.5.1. Localization and Spatial Reference System of the Geodata

Precise georeferencing is essential in all sensor-based precision agriculture studies [8]. All geodata (vector data: vines, raster data: DTM/DSM—Digital Terrain Model/Digital Surface Model, multi-channel raster orthomosaics) were referenced to Coordinate Reference System (CRS) EPSG (European Petroleum Survey Group) code 25,832 ETRS (European Terrestrial Reference System) 89/UTM (Universal Transverse Mercator) zone 32N. As further geolocation equipment, a ground station (DJI D-RTK Real Time Kinematic) and 2 Mobile Station GNSS (Global Navigation Satellite System) reference stations were used for absolute geolocalization of the drone and camera position during the flight. Additionally, several Ground Control Points (GCPs) were installed and precisely located with a portable Emlid Reach RS2- GPS (Global Positioning System) device to improve georeferencing, geometric rectification and reduce image distortions during postprocessing. The SAPOS (Satellite Positioning Service of German Land Surveying) guaranteed geometric corrections necessary for high-precision absolute location measurements. The open-source GIS software QGIS 3.22 (Quantum GIS, QGIS Development Team, Open Source Geospatial Foundation) was used for geodata visualization and all cartographic layouts.

#### 2.5.2. UAV Sensor Data Acquisition

Multispectral camera image data were recorded on five distinct spectral bandwidths (Red, Green, Blue, Red-Edge, Near Infrared) using the UAV Black Snapper XL Pro (quadrocopter) equipped with a MicaSense Red-Edge-MX^TM^ (Micasense Inc., Seattle, WA, USA) multispectral camera. An overview of the spectral bandwidth of each spectral band is summarized in Table 2. A calibration panel (Micasense Inc., Seattle, WA, USA) with known reflectance values was used for radiometric calibration. In agricultural application scenarios, multispectral sensors were often used for crop yield predictions [63,64], but also in more specific use-cases, like stress detection of vines [6,65,66]. Moreover, a LiDAR scanner integrated into a DJI Zenmuse L1 (RGB camera incl. gimbal) was used during UAV-sensor data acquisitions, installed on a DJI Matrice 300 RTK UAV, to generate point clouds, necessary for the generation of the DTM and DSM. 

The flight path for UAV- sensor data acquisition was created in block shapes within the flight planning software DJI flight Planner. The recording altitude above ground was set at 40 m, and the overlap for the recorded sensor images was set at 80%. The flight was accomplished midday with preferably cloud-free sky and perpendicular sunlight conditions to keep radiometric interferences as low as possible. After each flight, the recorded sensor image data were checked and saved on a prepared external hard drive.

#### 2.5.3. Photogrammetry

The photogrammetry software Agisoft Metashape Professional (version 1.7.0) was utilized in several processing steps to create the DSM, DTM, and multispectral orthomosaics [13] with a SfM (Structure from Motion)-based approach [9,67,68]) on the basis of the recorded images of the UAV- sensor data acquisitions (see previous Section). For this procedure, the imported images were adjusted, externally georeferenced, aerotriangulated, and readjusted. Georeferencing and image rectification were performed using GNSS coordinates of the five installed Ground Control Points (GCPs), matched in Agisoft Metashape professional version 2.1.4. Subsequently, the photogrammetric point cloud was derived to generate the multispectral orthomosaics, DSM and DTM. The orthomosaic, DTM, and DSM generation process involved three additional main steps: (1) initial processing for key point computation and image matching, (2) radiometric calibration for converting raw digital numbers into reflectance values, and (3) DSM generation for surface elevation representation and DTM generation through filtering of the ground classes. Finally, the orthomosaic was generated from the DSM and the calibrated spectral information. The georeferenced raster images were saved as TIFF (Tagged Image File Format) files (GeoTiff) and exported to QGIS version 3.22 for further geospatial processing.

#### 2.5.4. Single Vine Geopositions

The grapevines’ geoposition were manually set, based on the high- precise RGB orthomosaics from canopy-free season generated from acquisitions during November 2021. To ensure a unique identifier for each grapevine stem, the vines were systematically numbered by each row from west to east and north to south in slope line, from top of the slope to foot of the slope, to ensure systematic geographical sorting and unique identification. Progressively, during ground truth labeling the open-source geographical digitalization App QField was used to label the vines according to the growth classification assessment after [43]. Subsequently, the attributed point-shapefiles were re-exported to QGIS version 3.22. A map of the growth class and health condition labeled grapevines after Porten [43] is shown in Figure 4. The localized and ground truth labeled vines are the basis for all subsequent geoprocessing and ML classification procedures (Section 2.6 and Section 2.7).

### 2.6. Geoprocessing Workflow

Figure 5 summarizes the methods and workflow for post UAV- data acquisition geoprocessing and iterations. The processing workflow was mainly developed in QGIS version 3.22 (PyQGIS) and SAGA-GIS version 7.8.2, with integrated geoprocessing tools like GDAL (Geospatial Data Abstraction Library). Different authors emphasized the value of GIS and geospatial tools for agriculture and viticulture [39,69,70,71]. The following Section 2.6.1,Section 2.6.2,Section 2.6.3,Section 2.6.4,Section 2.6.5,Section 2.6.6,Section 2.6.7,Section 2.6.8,Section 2.6.9 and Section 2.6.10 describe the multiple geoprocessing steps in more detail.

#### 2.6.1. Vine Row Mask for Feature Extraction

QGIS 3.22, SAGA- GIS 7.8.2, together with the Python API (PyQGIS), were used to calculate the spectral features (VI), structural features (e.g. CHM), and texture features (GLCM). Image processing and segmentation were tested and performed using SAGA- GIS 7.8.2 and Python (libraries scikit-image). The usefulness of differentiating image features into spectral, structural, and textural features is reviewed in [39] and applied, for example, to oat yield predictions [33] or vine yield predictions [35]. The specific spectral, structural, and texture features were selected according to studies in agri- and viticulture combined with correlation analysis in accordance to the growth class ground truth.

The following subsections describe the generation of the different feature types. For the spectral features, those often used in viticulture context were selected. In the case of the structural and texture features, those generally applied for crops, biomass, and vegetation volume predictions were chosen. The general assumption was that spectral, structural, and texture features used in agriculture and, more specifically, in viticulture studies should be suitable as classification input for machine learning models concerning growth class and vigour predictions.

#### 2.6.2. Spectral Feature Type for Classification (Vegetation Indices)

Threshold values of VI for vine row extraction were set and compared to each other to select the best extraction result. In addition to standard Vegetation Indices like NDVI (Normalized Difference Vegetation Index), NDREI (Normalized Difference Red Edge Index), OSAVI (Optimized Soil Adjusted Vegetation Index), and TSAVI (Transformed Soil Adjusted Vegetation Index), more unusual indices such as the NDWI (Normalized Difference Water Index) or GNDVI (Green Normalized Vegetation Index) were also calculated for correlation with the growth classes and as potential feature inputs for the machine learning models (see Table 3). All VI were calculated in QGIS 3.22 with the native raster calculator using the spectral bands of the multispectral orthomosaic. The VI were mainly selected based on the comprehensive review of [38] and their different application fields, primarily biomass, yield, vine row extraction, or water stress detection and classification. 

#### 2.6.3. Structural Feature Types for Classification

The CHM was calculated as the difference between DSM and DTM (DSM-DTM). The CHM and grapevine geolocations were subsequently exported to QGIS version 3.22 /ArcGIS (ArcMap version 10.5) to calculate structural height features around each grapevine stem, including average height (H_mean_), maximum height (H_max_), minimum height (H_min_), median height (H_median_), and height standard deviation (H_std_) (see Table 4). This was done as part of the zonal statistic spatial aggregation process (see Section 2.6.9). Moreover, the CHM volume was calculated and later used as additional structural input feature for ML growth class prediction. The derived structural features were also correlated with the spectral features, texture features, and the growth class ground truth, to identify the most important parameters for model input feature selection.

#### 2.6.4. Texture Feature Types for Classification

Texture features were computed using the GLCM texture algorithm, introduced by Haralick et al. [78] and applied by several authors [33] in agricultural application fields. Basicallly, the GLCM provides information on the spatial relationship of pixel pairs in an image and is widely for image texture- based feature extraction and analysis in remote sensing. Using SAGA- GIS 7.8.2, different textural features were calculated for each of the five bands (R-G-B-RE-NIR), including variance (VAR), mean (ME) homogeneity (HOM), dissimilarity (DIS), contrast (CON), entropy (ENT), angular second moment (ASE), and correlation (COR). A 3 × 3 moving window was set for the calculations applied to each band of the multispectral orthomosaic. Due to the high correlation of the GCLM texture measures with the vegetation- sensitive spectral bands of the orthomosaic (R- NIR) the texture features of these channels were further investigated (correlation coefficient greater than 0.4 after Pearson [79]). Further details about these texture measures respectively features can be found in [80]. Table 5 presents the list of GLCM textural measures investigated in this study with the applied formula. The texture parameters were chosen as one more input feature group for the ML models to compare the model performance with the spectral feature group (VI) and the structural feature input group (e.g. CHM volume) and their combinations.

#### 2.6.5. Extraction Mask Generation

The extraction mask of the vine rows was created by fusing a pixel-based threshold mask with the a mask created by using an object-based segmentation approach. The generated combined mask was utilized to extract the vine row canopy and the associated spectral, structural, and texture image features necessary for ML model input and the classification process.

#### 2.6.6. Pixel-Based Mask

The OSAVI is especially useful for correcting spectral soil influences by optimizing the soil line (sl = 0.16). Various authors have discussed the Vegetation Indices mainly used for health monitoring and isolating the vine rows from the background primarily for extracting vine rows [81]. Under consideration of these studies, the OSAVI was selected from the VIs to generate the pixel-based mask (OSAVI mask). The OSAVI threshold (OSAVI > 0.7) was applied to create the binary OSAVI mask for vine row extraction in combination with visual validation. Alternatively, the threshold was determined using Otsus’s histogram-based method [82]. Moreover, threshold values from literature for vegetation, biomass, and crops were tested [83]. Compared with other selected VIs, OSAVI showed the best performance for vine row extraction and intermediate to high correlation with growth classes, according to the assessment of [43]. Technically, the mask was created using the raster calculator in QGIS 3.22.

#### 2.6.7. OBIS (Object-Based Image Segmentation)-Based Mask

The OBIS-based segmentation methods use a different approach compared to classical pixel-based methods: they group the pixels rather than classify each pixel separately with machine learning techniques [84,85].

For the object-based image (unsupervised) segmentation of the OSAVI image, the OBIA (Object-Based Image Analysis) algorithm after Adam and Bischof [85] with the neighborhood classification of Moore from the SAGA- GIS Toolbox (Image Analysis- Image Segmentation GUI (Graphical User Interface)) with k-means clustering postprocessing was applied. Furthermore, variance in feature space was set to zero, variance in position space was set to zero, the similarity threshold was set to zero, the generalization was set to 1, the number of clusters was set to 6, and the clustered split option was checked.

The technical approaches of the OBIA techniques included in SAGA-GIS are explained in detail in [86,87,88] and other related studies. This OBIA technique has been used in multiple remote sensing studies in which complex images have been segmented. According to the documentation, the SAGA- GIS methodology of OBIS (Object-Based Image Segmentation) was applied [87]. The two methods analyze cells on an image using an approach similar to the lattice network. The two approaches tested in this study include the Neumann approach with Neighborhood 4 and the Moore approach with Neighborhood 8. The algorithms determine the most effective object features in OBIA, ensuring high separability among landscape features [89]. The Moore neighborhood is one of these methods based on a 2D square lattice with a central cell and eight neighbor cells. Moreover, the algorithm allows four more cells as ‘neighbors of neighbors’ to be considered as the extended neighborhoods of cells. Using the neighborhood analysis approach approved for pixels on an image, the machine can quickly group pixels based on the similarity of the spectral brightness of their neighbors. Both methods of neighborhood analysis are well-known techniques used for analyzing the pixels in cell automata of Object-Based Image Segmentation [86]. Finally, the vine row mask was manually selected and rasterized for subsequent feature extraction and spatial aggregation.

#### 2.6.8. Merging Vine Row Masks

The final vine row mask was generated by fusing the pixel-based OSAVI mask (Section 2.6.6) with the OBIA mask (Section 2.6.7). The generated mask was utilized to extract the calculated spectral, structural, and GLCM texture features. These steps were technically achieved by using the native QGIS 3.22 raster calculator, which ensured that only vine canopy-exclusive pixels were considered for the spatial aggregation of spectral, structural, and textural features (following section). 

#### 2.6.9. Spatial Data Aggregation (Zonal Statistics)

The previously classified and extracted vine row pixels were subsequently used to aggregate the previously calculated features (spectral, structural, texture) using zonal statistics. In the process the mean, standard deviation, variance, minimum, maximum, range, and pixel count around each vines’ location were computed. Several studies have used zonal statistics for pixel volume calculation in different research fields of viticulture [28,90]. Technically, zonal statistics were performed in QGIS 3.22 with the Python API PyQGIS. The corresponding pixel values were aggregated within a rectangle measuring 1 m width (average width of vine rows) and a length of 2 × 0.60 m (1.20 m) (approx. average distance between individual vines in the same row for previous and subsequent grapevine stem in row) located around the individual vines (see red sampling rectangles around vines’ geopositions in Figure 6). The sampling rectangles were adjusted according to the average aspect of the slope derived from the DTM. For aggregation itself, only features intersecting the sampling rectangles and the vine row mask (see previous section) were used for spatial aggregation. The results of the zonal statistic aggregation were automatically joined onto the grapevine point shapefile as additional columns within the attribute table of the shapefile. The aggregated values around the single vines were used to calculate the CHM volume (see Section 2.6.10). Most importantly, the prepared vine canopy extracted spectral, structural and texture features for each grapevine, presented as one row in the attribute table, were now ready as input for the ML-based growth classifications.

#### 2.6.10. Calculation of the Canopy Height Model (CHM) Volume for Single Vines

The CHM-based pixel volume of the single vines was calculated by combining the CHM with the different extraction masks (extracted features) to consider only pixels previously detected as part of the vine row. The CHM was calculated as the difference between the DSM and the DTM within the vine row mask. Subsequently, the volume (CHM volume) of the individual vines was estimated from the mask’s product-extracted CHM with the area and number of vine row pixels within the zonal statistics sampling rectangles intersecting the vine row mask around each vine. Due to its high correlation with the growth classes, the calculated single vine volume was chosen as an additional input parameter for the model training (see Table 6). The Mann–Whitney-U-test was used to quantify the separability of the growth classes after [43] based on the CHM volume for each growth class combination pair. Similar approaches have been applied to estimate the pixel volume of vine rows and stems [28]. Moreover, based on the overlapping of the canopy of adjacent vines, the total real canopy volume of each vine cannot be 100% precisely computed. Therefore, the calculated values have a variance error due to this overlapping effect besides other methodological errors that sum up to the total uncertainty of the absolute vine volume estimation.

### 2.7. Growth Class Estimation Modeling

Following the geoprocessing and image processing steps (Section 2.6.1,Section 2.6.2,Section 2.6.3,Section 2.6.4,Section 2.6.5,Section 2.6.6,Section 2.6.7,Section 2.6.8,Section 2.6.9 and Section 2.6.10), the growth class prediction, based on two selected ML classifiers (RFC, SVM), was an essential step to derive overall and growth-class-specific prediction metrics to evaluate the suitability of the extracted features for predicting growth classes according to the assessment of [43]. For the ML classification process the Python library sklearn was used to fit the training data to the ground truth target classes with the ML classiifers. Before model fitting, the feature groups (spectral, structural, and texture) and their combinations (1–7) (see Table 6) were selected, scaled (standard scaling), and cleaned for NoData values. 

The generation of the test and training datasets of the input features were performed with repeated-k-fold- cross-validation (repeats = 5, splits = 5, random state = 1, n jobs = −1) by splitting the input data (input features and growth class ground truth) into train and test data sets. From GLCM derived features, those with the highest correlation after Pearson (calculated with RStudio version 4.3.3), to the growth classes were selected as input features in addition to the spectral and structural features. Mostly band 3 (red band) features were included in this category due to the high correlation between the red band and vegetation vigour. The RFC and SVM classifiers were chosen to fit the training data to the growth class ground truth data. These classifiers belong to the most efficient classification methods in agriculture crop and biomass prediction, and detection, including grapevine yield estimations [8]. Subsequently, the hyperparameters of the models were optimized with a hyperparameter-tuned grid search [33]. The essential steps for the model generation are summarized in Table 7. The model’s output generated n = 25 single metric outputs for each growth class. These values were used for univariate statistics and the evaluation of the consistency of the metrics’ output. For model evaluation, the validation metrics were calculated for the train and test data to search for overfitting and underfitting tendencies. The models and the validation metrics (accuracy, f1-weighted) were selected based on literature sources, primarily, biomass estimation and crop modeling research. Nevertheless, the specific geometrical properties of vine canopies and class frequency balance of the label dataset were also considered by several authors [91,92,93], adapting the classification procedure accordingly. These aspects are also shortly considered in the discussion section (see Section 4.4). Furthermore, the f1-weighted scores were calculated to test for the impact of imbalance on the generated dataset [93,94,95]. To quantify the statistical difference of the average model prediction metric results, the Mann–Whitney-U-test was applied for all model combinations (1–7) and metric pairs (accuracy vs f1-weighted). Moreover, the class-specific accuracy (confusion matrix) was calculated for the model output with the best overall accuracy score. Additionally, user and producer accuracy were calculated. The user accuracy for each growth class was calculated as the ratio of the sum of all correctly identified growth classes for this class, divided by the sum of all other incorrectly labeled growth classes for the same growth class and expressed in %. For producer accuracy, for each growth class, the sum of all correctly identified growth classes for this growth class was divided by the sum of all other incorrectly (ML model) predicted growth classes for the same growth class and also expressed in %. This approach allowed for a more specific discussion of the classification results and trends of the individual growth classes (see Section 4.6). 

## 3. Results

The following subsections present the CHM volume calculation and the results of the growth class ML model predictions based on the geoprocessing workflow from Section 2.6.1,Section 2.6.2,Section 2.6.3,Section 2.6.4,Section 2.6.5,Section 2.6.6,Section 2.6.7,Section 2.6.8,Section 2.6.9,Section 2.6.10 and Section 2.7.

### 3.1. CHM Volume Results

The results of the CHM model calculation are shown Figure 7. The stars (*) connected by the brackets indicate the strength of significant differences between the growth class grouped CHM volume calculations, based on the paired Mann–Whitney-U-test (Figure 7). The graph shows that all neighbor growth classes significantly differ to each other according to the CHM volume. This CHM volume growth class separation is more pronounced than spectral feature only approaches (e.g. OSAVI). Subsequently, the CHM volume calculation result was used as one additional structural input feature for ML classifications. 

### 3.2. Overall Growth Class Prediction

In the following Section 3.2 the major results of the classification metrics are presented according to the different input feature types and their combinations.

#### 3.2.1. Spectral Feature Group Estimation

The best model performance for the spectral input feature group (1) of the test data set was received from the SVM 1 model with an accuracy = 40.65% and f1-weighted of 37.48.

On the other hand, the RF 1 model yielded the lowest accuracy (accuracy = 30.41%, f1-weighted = 20.98%). For the train data the best model performance for the spectral feature group was received from the SVM 1 model with an accuracy of 50.48% and f1-weighted of 46.76%. For the train dataset, the RF 1 model predicted the lowest accuracy (accuracy = 31.51%, f1-weighted = 22.25%) (Table 8).

#### 3.2.2. Structural-Feature-Based Growth Class Estimation

Higher estimation accuracies were obtained from models based on the structural input feature group (2). The best model performance for the structural feature group of the test data was received from the SVM 2 model with an accuracy = 47.38% and f1-weighted of 46.21%. On the other hand, the RF 2 model yielded an accuracy of 40.4% and f1-weighted of 31.57%. The best model performance for the structural input feature group of the train data was received from the SVM 1 model with an accuracy of 50.34% and f1-weighted of 51.23%. For the train dataset, the RF 2 model yielded the lowest accuracy (accuracy = 41.58%, f1-weighted = 32.66%) (compared in Table 8). The structural feature types influences prediction metrics more strongly than non-structural features, including models 1, 3, and 5. The higher bivariate correlation coefficients indicate the improved metrics scores compared to the spectral features (see Appendix A).

#### 3.2.3. Texture-Feature-Based Growth Class Estimation

The best model performance for the texture-feature-based group (3) of the test data was received from the SVM 3 model with an accuracy = 37.98% and f1-weighted of 47.72. On the other hand, the RF 1 model yielded an accuracy of 33.17% and f1-weighted of 28.32%. The best model performance for the train data texture input feature group was received from the SVM 1 model with an accuracy of 40.13% and f1-weighted of 52.49%. For the train dataset, the RF 3 model yielded an accuracy of 34.10% and f1-weighted of 29.10% (compared in Table 7). The texture feature group input predictions are less favorable than the structural feature inputs and are comparable to the spectral feature input results.

#### 3.2.4. Summary of Impact of Feature Types and Feature Type Combination on Model Classification

In comparison, model predictions show significant differences according to the chosen input feature types, feature group, and selected ML classifier. Structural features like CHM and the calculated CHM volume led to improved classification metrics compared to spectral features (Table 8). For the third texture-features-type only input group (3), model predictions were slightly better than for the spectral feature only input group (1) of RFC. In contrast, texture features they are even less favorable in the case of the SVM classifier compared to spectral features. Generally, the spectral- and texture-only feature groups differ only slightly and have the least desirable classification metrics for both classifiers. Moreover, combining all input feature groups and feature types leads to the best and most robust classification results for growth class prediction, also for both classifiers (RFC, SVM). Nevertheless, the RF classifier showed decreased prediction metrics for most input feature groups compared to the SVM classifier (Table 8). In summary, selecting and combining the extracted input features is essential for model prediction results concerning growth class predictions.

#### 3.2.5. Overall Growth Class Prediction

The metrics for the train and test dataset predictions for the repeated k-fold cross-validation (n = 25) ML classifications (1-7) are summarized in Table 8 and visualized for the accuracy of the SVM and RFC classifier models as boxplots with significance stars derived from the Mann–Whitney-U-test in Figure 8 and Figure 9.

ML models including the structural input feature group (2,4,6,7), are marked in gray to discuss the influence of the structural features compared to the other feature types in Section 4. The highest accuracy of the test data prediction with an accuracy = 48.51% and f1-weighted = 45.50% was obtained from the SVM classifier with the combination of spectral, structural, and texture features as input for model training fit (SVM 7). The highest accuracy of the train data prediction (accuracy = 53.7%, f1-weighted = 50.11%) was also obtained from SVM modeling with the combination of all feature type groups (SVM 7) as input features. For the RFC classifier, the highest test data accuracy (accuracy = 40.0%, f1-weighted = 31.5%) was also achieved with the combined spectral, structural, and texture feature input (RF 7), followed closely by the structural-feature input-only group accuracy (accuracy = 40.65%, f1-weighted = 37.5%) (RF 2). Regarding training data accuracy, the structural feature group (2) received the best accuracy of 41.58% and f1-weighted of 32.66% for the RFC classifier (RF 2). Moreover, all models containing structural feature inputs (2,4,6,7) have, on average, improved classification metrics compared to the non-structural-feature-containing models (1,3,5) for both classifiers (RFC, SVM). The worst prediction metrics are obtained from the RF classifier models, which contained no structural feature inputs for training (RF 1, R F 3, RF 5). Within the SVM classifier group, the models without structural features (SVM 1, SVM 3, SVM 5) showed decreased classification metrics compared to models with structural feature input (SVM 2, SVM 4, SVM 6, SVM 7). Generally, the SVM classifier obtained consistently higher accuracy and f1-weighted scores on the average order of 10- 20% of absolute accuracy. Moreover, the standard deviations for the metrics also showed less range for the models with the best model mean scores (SVM 4, SVM, 6, SVM 7) compared to the less successful prediction models (RF1, RF 3, RF5). Therefore, the SVM models, especially including the structural-feature-type group as input, seem more robust and consistent than the other RFC models without these feature types.

## 4. Discussion

The following section discusses the results presented in Section 3 within the context of vine row extraction, correlation of the growth classes to the calculated CHM pixel volume, and the classification metrics of the ML models.

### 4.1. Extraction Masks

The pixel-based and OBIA-based vine- row extraction approaches are compared and discussed in multiple remote sensing studies and reviews [10]. The general approach in this study used a combined pixel- and OBIA-generated mask for spectral, structural, and texture feature extraction. Moreover, the merged vine row mask metrics were evaluated visually and calculated for a short segment of a vine row according to the OBIA accuracy assessment from [96,97]. The intersection with the digitized ground truth vine row is approximately 89% for the combined mask, indicating a relatively good fit with the template mask. Nevertheless, the focus of this study was not to optimize the vine row mask; therefore, this aspect will not be discussed in more detail here.

### 4.2. Structural Parameters and CHM Volume

The results of the CHM volume calculation (Figure 7) reflect the high correlation and influencing power of the CHM volume for growth class separation. Other studies have also detected and discussed the high correlation power of structural features like the CHM or the CHM volume [10]. Moreover, these studies also showed a strong correlation between the CHM and the CHM volume for biomass and crop yield estimation in different agricultural scenarios. CHM models were primarily used for vine row extraction. Moreover, several authors used the CHM, extracted pixels for volume calculation, and compared them with ground truth measurements [81]. Several authors discussed the structural parameters’ power as prediction parameters and input features for other regression analyses and classifications, especially ML-based classifiers like SVM or RF [33]. In most cases, structural parameters showed superior biomass and growth prediction ability compared to spectral features. This is also indicated by the high correlation coefficients and significance tests, which revealed more vital effect sizes for CHM volume than spectral features like the OSAVI against the growth classes. A high correlation between the vine’s volume, biomass yield, and structural parameters (CHM) is discussed by several authors [10,28,98]. Nevertheless, due to the complex internal structure of grapevines, the CHM and also manual measurement methods can only approximate the actual volume of vines [3]. The influence of structural parameters on machine learning model classification in this study is further discussed in Section 4.4.2. Due to the highly positive correlation of the CHM volume to the growth classes after [43], the CHM volume is used as an additional input feature for ML model training.

### 4.3. Model Validation

Evaluating ML model predictions is essential for interpreting classification metrics, robustness, and reproducibility of the growth class prediction [92]. Comparing the generally higher training to test metric score results hints at overfitting of the model predictions due to the limited amount of available training data, including ground truth data, multispectral orthomosaics, and CHMs during sampling time (compare train and test data metrics from Table 8 and Figure 10). Especially the SVM models show a higher difference between the train and test data (higher train than test score) values of the repeated cross-fold validated overall accuracy (OA), which is probably a result of the sensitivity of kernel-based classification models. The tendency of overfitting for regression, machine learning, and deep learning models was also observed by [99].

Moreover, to test for imbalance, the OA is compared with the f1-weighted score (Figure 11 and Figure 12). The f1-weighted score considers the sample size for each growth class about the total sample size. Pronounced differences between the overall accuracy and the f1-weighted score indicates an imbalance of the dataset due to the underrepresentation of some growth classes (e.g. growth classes 0, 1, 9) compared to more frequent growth classes (e.g. 3, 5, 7). This issue is also considered in the discussion part of Section 4.6. In ongoing studies, more training data from multiple sampling dates and different vine fields should be incorporated to improve the model predictions’ fit, consistency, and reproducibility. Moreover, more samples from less frequent growth classes should be integrated to avoid imbalance bias in the prediction metrics. Therefore, a batch workflow to calibrate and process the data from different data acquisitions should be developed in future studies.

### 4.4. Model Performance Evaluation

The impact and contribution of the essential feature types and input feature groups is evaluated in the following section, considering their hierarchy in classifying vine growth and vine vigour.

#### 4.4.1. Influence of Spectral Features

Spectral features’ correlation and classification power for predicting biomass production and crop yield has been observed, with varying correlation strength considering the structure, growth characteristics of vegetation type, sensor type, and applied spectral indices [35,67,100,101,102,103,104,105,106,107,108,109,110,111,112,113,114,115,116]. Generally, the spectral feature type based classification was less successful, compared to models with structural feature type inputs (Table 8). This indication can result from the intermediate correlation of the growth classes with spectral features due to saturation effects and the complex internal geometric structure of the vine canopy [35]. Moreover, the SVM models predictions show improved classification metrics than the RF classifier with the same input features. However, spectral features are essential for predicting different types and origins of photosynthetic materials. Many studies used spectral features as input for yield predictions in various agriculture and viticulture application scenarios [64,116,117,118]. It is worth noting that models based on spectral features underestimated biomass samples with higher values. One reason for this could be optical saturation effects of the VI. Similar observations were observed in winter cover crop biomass estimation [119] and soybean biomass and LAI estimation [120]. Especially, VI based on NIR-band and red-band ratios (e.g. NDVI) tend to saturate for highly dense canopies [121,122], which resulted in the poor performance of predictive models due to saturation effects. Prabhakara et al. [119] reported that the NDVI showed asymptotic saturation in the higher range of rye biomass (>1500 kg/ha). VI are also environment- and sensor type-sensitive and -specific [123]. The surface sensor data derived from nadir’s view often do not reflect internal 3D canopy structure and geometrical patterns, depending on the crop or plant type. These factors distort the correlation between the growth class ground truth and the “real” volume of vines. Due to the complex internal architecture of vines [31,43], these effects are relatively pronounced for sensor- based viticultural investigations.

#### 4.4.2. Influence of Structural Features 

Many studies have validated the potential of structural features in biomass estimation [124,125,126] and vine row volume calculation [10]. Bendig et al. [127] reported that canopy height derived from the Crop Surface Model (CSM) is a suitable indicator of biomass in barley. Moreover, Shu et al. [128] used the CHM and the canopy coverage to estimate maize above-ground biomass. In this study, the structural input features provided increased classification metrics for both classifiers (RFC and SVM) compared to the spectral feature type-only model results (see Table 8). A possible explanation of the superior prediction performance based on structural features over those based on spectral features is that canopy structural features can provide more information about the three-dimensional canopy structure and, therefore, also show higher correlations to the growth classes and the CHM volume. The superiority in estimating biomass based on structural or combined feature types is also claimed by the study of Michez et al. [98] by combining LiDAR-derived height measures with VI.

#### 4.4.3. Influence of Texture Features

Several authors used texture canopy features and related vegetation components as input for biomass estimation or yield prediction of different crop types [10,33,39,129]. Delenne et al. [18] presented, developed, and discussed different texture extraction techniques for vine row detection. Therefore, texture features can potentially add explanatory power in characterizing the growth of vines besides grapevines’ spectral and structural signature [130,131]. Wengert et al. [132] recognized that GLCM-based textural features improved the estimation of barley dry biomass and Leaf Area Index (LAI). Similar results were also documented in the above-ground biomass estimation of legume grass mixtures [111], rice [105], and winter wheat [106]. A possible explanation for the improved estimation accuracy in some of these studies is that texture features consider the spatial variation in the pixels and provide additional information about the physical structure of canopies, including canopy edges, and the overall canopy architecture [123,133]. Nevertheless, in this study, the growth class prediction results are comparable to the spectral-feature-input-only predictions and inferior to the structural input feature models (see Table 8). This is probably a reflection of the same issue mentioned with the spectral features, capturing only surface nadir reflectance properties of the vine canopies, without capturing the complex internal structure. Moreover, texture features are based on spectral bands; thus, they show some collinearity with spectral features. However, in contrast to spectral features, texture features can characterize canopy architecture and structure patterns to some extent [130], as well as weaken saturation issues and suppress soil background effects. The only light difference between the spectral and texture features in this study does not hint at a superior performance of textural features over spectral features, like in the study of [131], where forest biomass was predicted. This non-agreement is probably a result of different vegetation types and associated structure, which play a significant role in sensor data correlation and varying types of prediction variables (continuous biomass yield vs. categorical growth classes). This may also reflect the texture features’ inability to consider minor differences in the canopy texture between the growth classes.

#### 4.4.4. Combining Different Feature Types

Numerous studies have highlighted the potential of combining structural features like CHM with spectral (VI) or texture features rather than using them separately [64,106,115,130,131,134,135,136,137,138,139,140], which resulted in robust and improved estimations of biomass yield. Moreover, [137,138,139,140] have tested the combination of multiple information sources by integrating non-spectral features (3D thermal sensors) or textural features with spectral features for evaluating grassland biomass and cultivated crops. Consistent with previous studies, the results in this work also show that a combination of spectral, structural, and texture features (7) resulted in improved accuracies and f1-weighted scores than spectral or structural features alone (see Table 8). Primarily, canopy structural features can provide valuable information about canopy architecture not supplied by spectral features and can, to some extent, overcome the saturation problem of spectral features. Nevertheless, the highly varying porosity of the canopy cannot be perfectly modeled based on this 2D assumption rather than an overall estimation. For vines, Matese and Di Gennaro [35] used spectral and structural input features derived from UAV images of vine fields as input for yield prediction. Compared to classical regression methods, they showed the advantage of combining spectral features (NDVI) with structural canopy features and the increased prediction metrics of machine learning classifiers, including RF, SVM, and GPR (Gaussian Process Regressor). Apparently, in this study, combining different input feature groups mainly including structural features (e.g. Canopy Height, Canopy Surface) leads to the best prediction results and most robust models with relatively low standard deviations for the growth classes after [43]. Shu et al. [99] also concluded that combining spectral, structural, and texture features provides superior prediction metrics compared to classical regression methods like PLSR (Partial Least Square Regression).

### 4.5. Comparison of Machine Learning Classifiers (RFC vs. SVM)

The classification results and trends based on the input features are similar for both classifiers (RFC, SVM), with improved prediction metrics for models with structural features that are exclusive (2) or a combination of features, including structural feature types (4, 6, 7) (Table 8). Generally, the SVM models showed slightly higher accuracies and f1-weighted scores than the RFC model predictions (higher accuracy, f1-weighted, lower Accuracy_std_ and f1-weighted_std_). This lower prediction quality is probably a result of the better performance of SVM when distinct classes are predicted, the sample size of the input data is relatively small, and the correlation between the input features and the growth classes is primarily linear. Moreover, the data size of the input features for each growth class could be too small, especially for the RF classifier, which generally requires more training samples for a sufficient model fit than SVM. Moreover, SVM could be more suitable due to the linearity of the growth classes and the margin optimization of the classes relative to each other. Otherwise SVM needs more time for model fitting, and it is more challenging to handle the hyperparameter tuning compared to RFC. Moreover, SVM is better at handling fewer samples with high input dimensions than RF, whereas RF is particularly effective and precise for many samples with high dimensions and target classes. Adugna et al. [141] discuss such discrepancies for the model classifiers, especially for the SVM and RF classifiers. Sheykhmousa et al. [142] compared RFC and SVM classifiers in the remote sensing applications of landscapes, considering many external factors like image resolution, sensor type, type of target classes, complexity of the landscape, and internal characteristics of the classifier, like, hyperparameter optimization such as kernel selection and number of training features for model fitting, influencing the prediction quality and metrics. Therefore, including more growth class samples as training input and widening the feature space could shift the more favorable classifier from SVM to RFC in future classifications, with extended datasets. A discussion comparing the prediction results and model robustness of different machine learning classifiers is also provided recently by [143].

### 4.6. Class-Specific Accuracies of the Best Model

To discuss growth class-specific prediction matches, a confusion matrix was calculated for the ML model with the best overall accuracy concerning the test data (SVM 7) (see Table 9). The pattern of the class-specific accuracy is similar for all models and classifiers, differing mostly in relative accuracy values due to feature inputs, classifier type, and selected hyperparameter. For discussing the class-specific accuracies, the confusion matrix with the highest overall accuracy was calculated (SVM 7, compare Table 8).

Class-specific accuracies indicate a more or less pronounced range of accuracy predictions for all growth classes, considering direct neighbor classes as matches, with overall prediction accuracies of more than 80% for most growth classes (see color code marks in the confusion matrix, Table 9). Therefore, strongly pronounced prediction outliers (e.g. labeled growth class 3 predicted as growth class 7 (5.01%)), or even higher discrepancies between ground truth and model prediction are rare or even not apparent (e.g. labeled growth class 9 predicted as growth class 3). This indicates the general suitability of the selected input feature groups for correlation and classification concerning the growth class assessment after [43]. The most substantial overlap with neighbor growth classes occurs for growth class 9 and growth class 0. This is probably a result of different factors, including more robust labeling errors, sensor data effects, or separation issues of the interrow area from the vine row section (especially growth class 0 and 1). A visualization of the difference between the growth class ground truth and model prediction for each grapevine position is presented in Figure 13. Especially, the intermediate growth classes 5 (67.7%) and 7 (57.1%) showed the highest match rate, which may result from less overlap and interference with vine row vegetation and soil background compared to the lower growth class spectrum (growth classes 0, 1, 3), on the one hand and fewer saturation effects compared to the highest growth class 9, on the other hand. Other reasons might be the varying number of ground truth samples for each growth class (imbalance of the dataset) combined with ground truth labeling errors. Moreover, differences in user and producer accuracies in Table 9 and the train and test data predictions (higher train than test data accuracy) indicate model overfitting. The difference in user and producer accuracy also hints at a lack of sensitivity and specificity compared to positive and negative precision (compare [144]). Therefore, user accuracy (indicating better training data prediction) provides slightly higher matches when summed up for all classes than producer accuracy (lower test data prediction), which is also indicated in Figure 10, for the overall prediction metrics.

On the one hand, the model predictions overestimate the low-growth classes (0,1), which results in negative values, most likely resulting from difficulties separating the vine row from interrow vegetation. Therefore, a substantial overlap with growth classes 1 and 3 is apparent. On the other hand, higher end-growth classes (7,9) are, on average, more underestimated by model predictions (see Table 9). This indication is probably a result of different effects and error sources, including saturation effects due to the complex internal structure of the vines, sensor view direction, sensor type, and labeling errors of the ground truth data. Such effects have also been observed in other remote sensing studies, like those of Zhen et al. [145]. Moreover, landscape features, including different grapevine varieties and vine stages, are more challenging to differentiate when structurally similar to each other, concerning the sensor data image features (spectral, structural, and texture features) [146,147]. Despite these limitations, the growth class assessment after [43] is basically suited for ML classification based on UAV-sensor data. The shortcomings are mostly the sum of different interacting factors mentioned above (labeling errors, sensor type, sensor direction, feature types, distortions), which could be overcome with other sensor types, respectively, with fusion of different sensor types combined with a reduction in ground truth labeling errors. Moreover, the training dataset size could be enhanced by using more ground truth and sensor data to improve the model fitting process and prevent an imbalance of the dataset. Progressively, additional and relevant input features like other ground truth data, such as infection type and strength, could improve the growth class predictions and be compared with sensor-data-only feature-based classification results. These results could be compared with similar sensor-derived feature inputs and used to validate ground truth transferability to sensor-derived feature classification. Lastly, if necessary, the assessment could be slightly modified to improve the classification results.

## 5. Conclusions and Future Work

In this study a new geoprocessing workflow was developed and evaluated in combination with ML model predictions (RFC, SVM) for separating growth classes according to the growth class assessment after Porten [43], based on different input (spectral, structural, texture) features derived from highly-precise georeferenced UAV-sensor data. A vine row mask was generated by combining pixel- and object-based image segmentation approaches for vine canopy extraction.

The selected ML classifiers (SVM, RFC) were relatively efficient in vine vigour classification. SVM produced higher estimation accuracies than RFC models. Therefore, integrating image feature types, like spectral, structural, and textural features, offers a promising avenue for general growth class prediction 

The primary outcomes of this study are as follows:(1)Combining spectral, structural, and texture features indicates the best classification results for both machine learning classifiers (Random Forest Classifier and Support Vector Machine Classifier). Nevertheless, SVM performed better than RFC, due to classifier properties and ground truth data set characteristics.(2)The structural input features are the most positively influencing feature type. The canopy structural features achieved higher accuracy and f1-weighted scores, than models using spectral or textural features alone or combined. Canopy structural features most likely provide a more accurate representation of canopy architecture than spectral and texture features.(3)Although less influential than structural features, canopy spectral and texture features are also an essential indicator for growth class estimation. They may also offer a means to overcome saturation issues associated with spectral features from nadir camera positions.(4)The class-specific accuracies show that most growth classes were correctly predicted, or the mismatch between the predicted and labeled classes was only minor. Therefore, the class-specific accuracy would increase to over 80 or 90% when considering the neighborhood growth classes as matches. Nevertheless, some classes, like growth class 0, 1, or 9, are difficult to separate with sufficient probability due to different error sources and distortions. These include subjective ground truth labeling (inter- and intrapersonal sources of error), uncertainty in distinguishing the growth classes, different approaches of the individual classification methods, complex correlation and varying influence strength of input parameters to the particular ML growth class prediction, and other disruptive influences (e.g. radiometric interference) during data acquisition.(5)The comparison of training and test datasets overall accuracy and growth class specific user and producer accuracies of the class-specific prediction hint to overfitting of the different models for both classifiers (SVM, RFC). Moreover, the difference between unbalanced (accuracy) and balanced prediction metrics (f1-weighted score) indicate imbalance of the ground truth dataset, which should be further considered and improved in future studies, for example, by integrating ground truth data from other years, growth stages and other vine fields.

In summary, UAV-derived multispectral image data can provide a good foundation for growth class and grapevine vigour predictions. Nevertheless, there are several limitations to this approach. Based on the complex geometry of vine canopies, 2D multispectral and CHM approaches oversimplify the complex 3D geometrical structure of grapevine canopies. Therefore, considering additional sensor types like LiDAR-sensors combined with multi-, hyper-, and thermal sensor data provide additional geometrical, spectral and thermal information. Combined with enhanced sensor data acquisition and subsequent processing routines, extended sensor data types and sensor data fusion techniques can improve growth class and vine vigour predictions even further. Hence, future work can contribute to enhance growth classification assessments, vigour predictions and canopy volume estimations, for a more efficient vineyard management and viticulture automatization approaches.

## Figures and Tables

**Figure 1 sensors-25-00431-f001:**
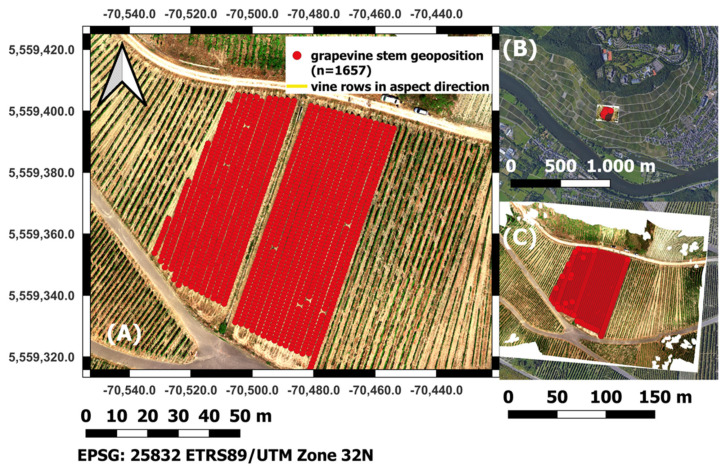
(**A**) Investigation area in Bernkastel-Kues within the Moselle wine region mapped on a high-precision orthomosaic (CRS (Coordinate Reference System) with EPSG (European Petroleum Survey Group) 25,832 ETRS (European Terrestrial Reference System) 89/UTM (Universal Transverse Mercator) zone 32N). Red points represent each vine position that was localized with differential-GPS (see Section 2.5 for more information) (**B**) Zoomed out view of the investigation area and overview of local vineyard structure and the Moselle river (mapped on Google Earth Satellite map from QuickMapService plugin in QGIS version 3.22) (**C**) Intermediate zoom of the investigation area, with orthomosaic on Google Earth Satellite map. It can be seen that the UAV- sensor- based orthomosaic and the Google Satellite map show some offset to each other, due to different absolute geographic accuracies and spatial resolution.

**Figure 2 sensors-25-00431-f002:**
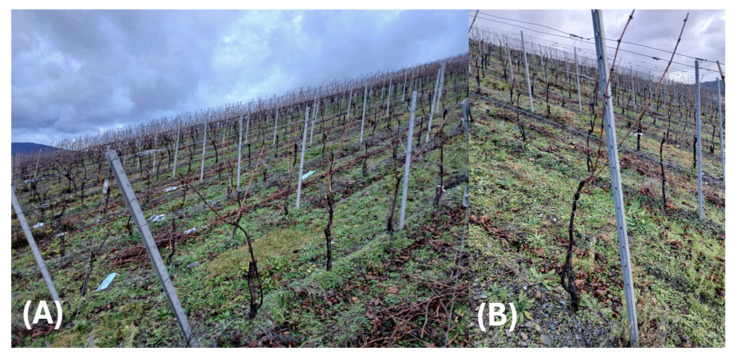
(**A**) Zoomed-out view of the canopy-free vine rows of the investigation area. (**B**) Zoomed-in view of the training system in the investigation area (photos taken in December 2024).

**Figure 3 sensors-25-00431-f003:**
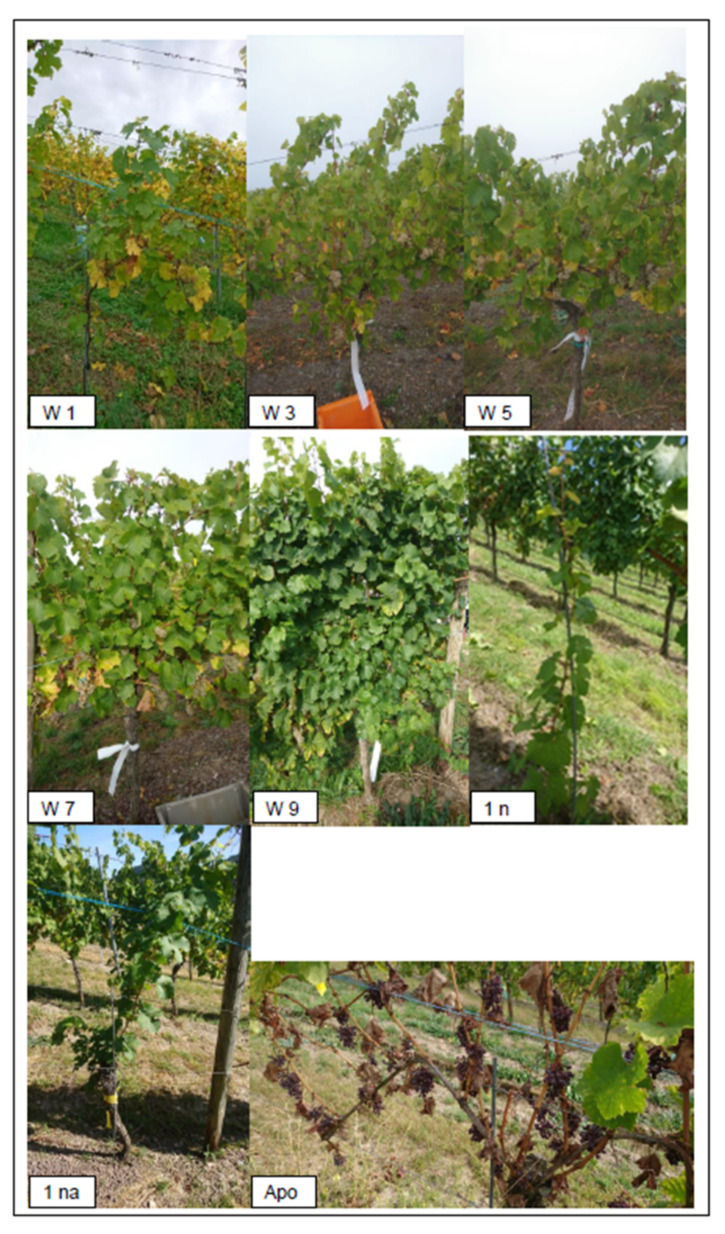
Ground truth template examples with label descriptions for the growth classes after Porten [43]. The specific visual characteristics and correlations to viticultural, oenological, and environmental parameters are described in detail by [43].

**Figure 4 sensors-25-00431-f004:**
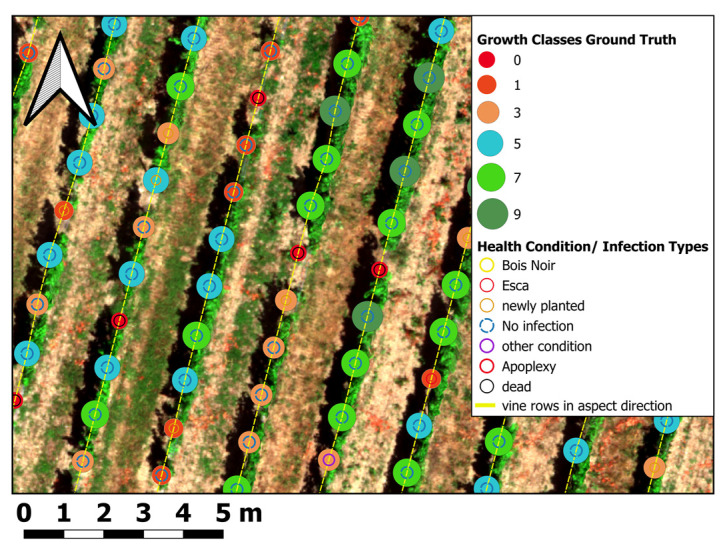
Color-coded growth classification after Porten [43] for single grapevines in the investigation area mapped on multispectral orthomosaic. All geodata are projected to CRS with EPSG: 25,832 ETRS89/UTM zone 32N.

**Figure 5 sensors-25-00431-f005:**
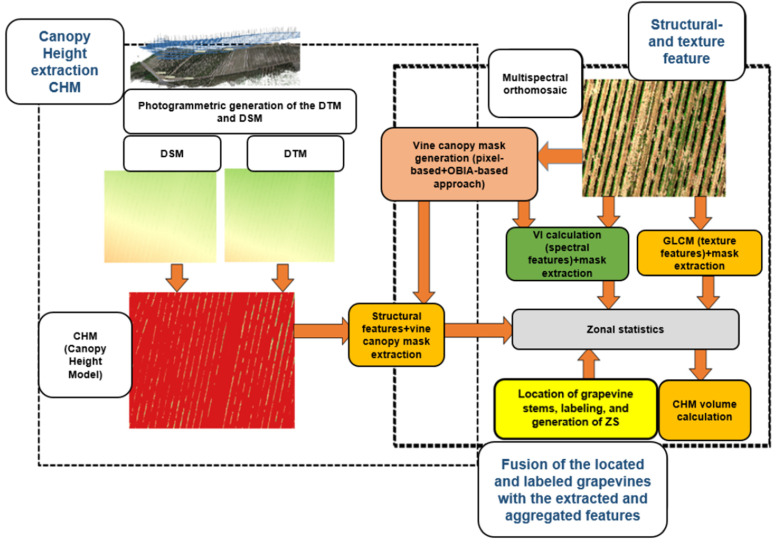
Visualization of the developed and applied geo- and image processing workflow in this study, with QGIS and different geospatial libraries in Phyton. Geoprocessing was the foundation for further statistical analysis, and machine learning model predictions of the growth classes after [43].

**Figure 6 sensors-25-00431-f006:**
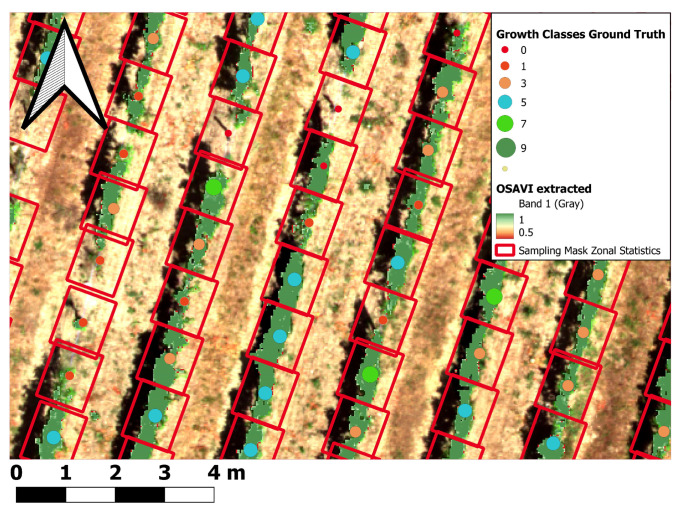
Sampling rectangles around the vines’ position for zonal statistics pixel aggregation process, together with growth class categorized grapevine stem positions and vine row extracted OSAVI (OSAVI extracted). All geodata are projected to CRS with EPSG: 25,832 ETRS89/UTM zone 32N.

**Figure 7 sensors-25-00431-f007:**
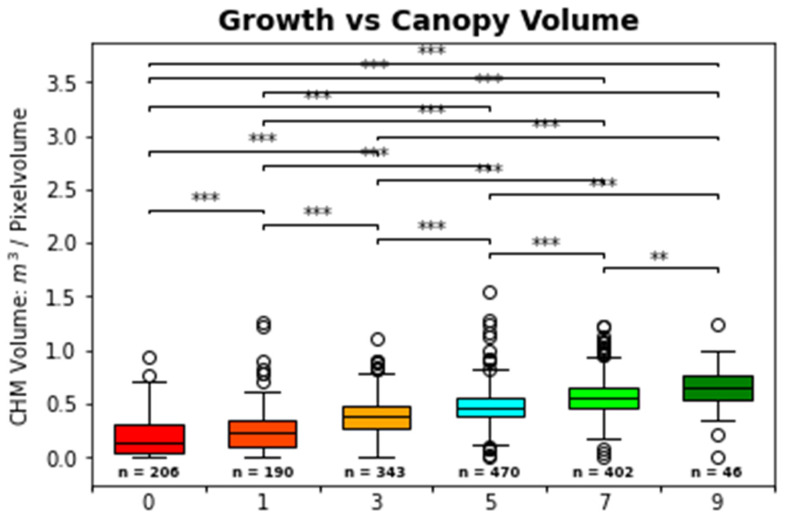
Growth class grouped CHM Volume boxplots with significance stars (*) between boxplots generated according to the Mann–Whitney-U-test with *p*-value significance. ** Signal greater than 0.01 (intermediate significance). *** Signal less than 0.001 (vital significance).

**Figure 8 sensors-25-00431-f008:**
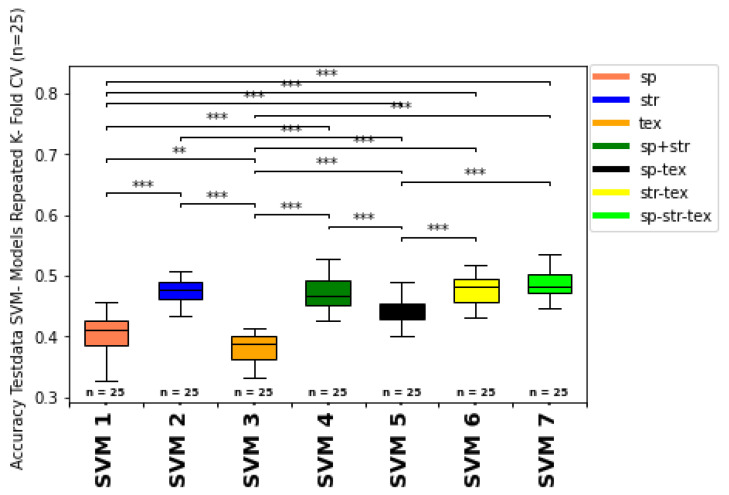
Input feature group (1–7) grouped boxplot OA (overall accuracy) in % for the SVM classifier of the seven different SVM models (see legend color of grouped boxplots), with significance stars (*) generated according to the Mann–Whitney-U-test between statistical significant model (SVM 1–SVM 7) results, where significant accuracy differences, derived from repeated-k-fold-cross-validation occurred, with *p*-value significance classes: ** Signal greater than 0.01 (intermediate significance). *** Signal less than 0.001 (vital significance). No stars between boxplots indicate no statistical differences between the model outputs according to Mann-Whitney-U-test.

**Figure 9 sensors-25-00431-f009:**
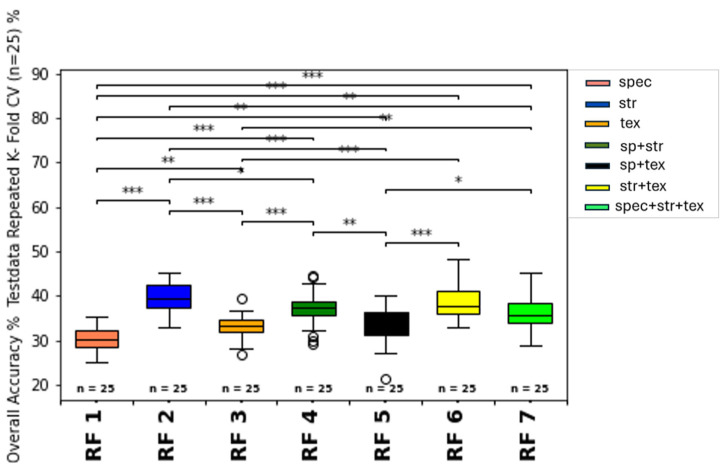
Input feature group (1–7) grouped boxplot accuracy in % for the RF classifier of the seven different RF classifier models (see legend color of grouped boxplots), with significance stars (*) generated according to the Mann–Whitney-U-test between statistical significant model (RF 1–RF 7) results, where significant OA (overall accuracy) differences derived from repeated-k-fold-cross-validation occurred, with *p*-value significance classes: * Signal greater than 0.1 (weak significance).** Signal greater than 0.01 (intermediate significance). *** Signal less than 0.001 (vital significance). No stars between boxplots indicate no statistical differences between the model outputs according to Mann-Whitney-U-test.

**Figure 10 sensors-25-00431-f010:**
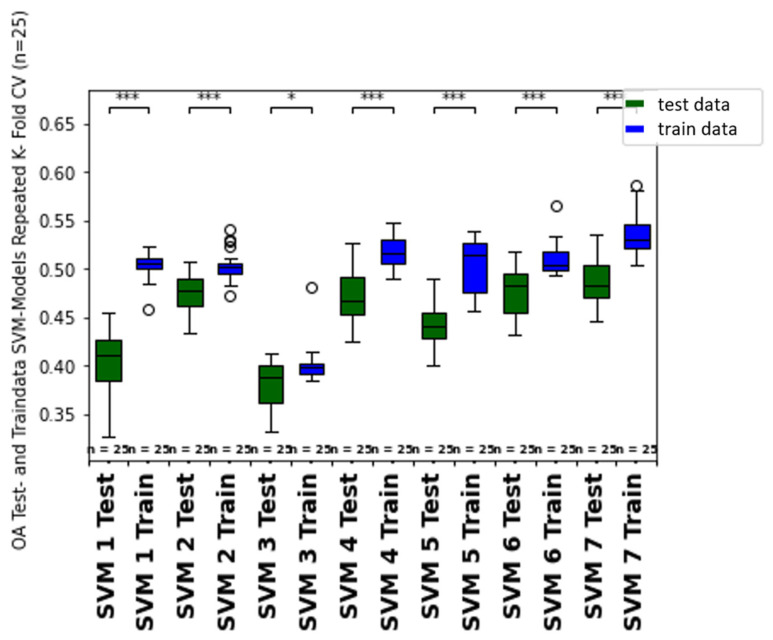
Pairwise statistical comparison of OA (overall accuracy) of the test and train data in % for the SVM classifier of the seven different feature groups input sets (1–7) with significance stars (*) generated according to the Mann–Whitney-U-test between boxplots where significant accuracy differences (OA) derived from repeated-k-fold-cross-validation occurred, with *p*-value significance classes. * Signal greater than 0.1 (weak significance). *** Signal less than 0.001 (vital significance).

**Figure 11 sensors-25-00431-f011:**
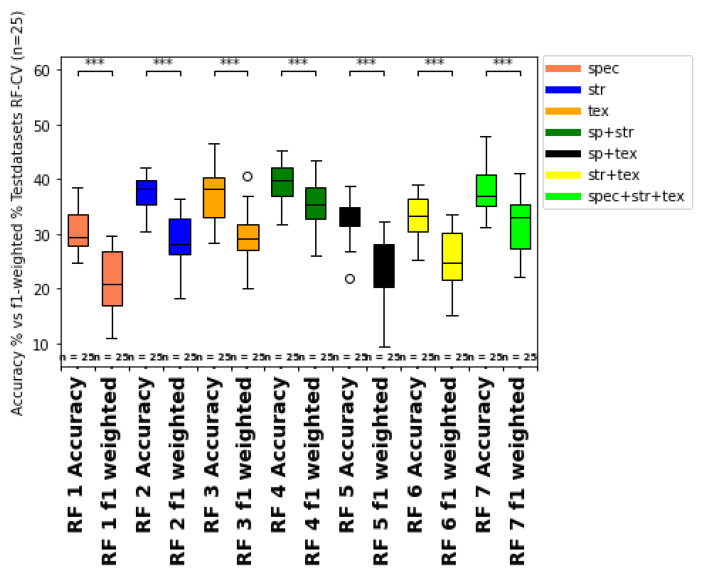
Pairwise statistical comparison of accuracy in % and f1-weighted score in % of the test data sets for the RF classifier of the seven different input feature groups (see legend color of grouped boxplots) with significance stars (*) generated according to the Mann–Whitney-U-test between accuracy and f1-weighted, with significant differences derived from repeated-k-fold-cross-validation occurred, with *p*-value significance classes. *** Signal less than 0.001 (vital significance).

**Figure 12 sensors-25-00431-f012:**
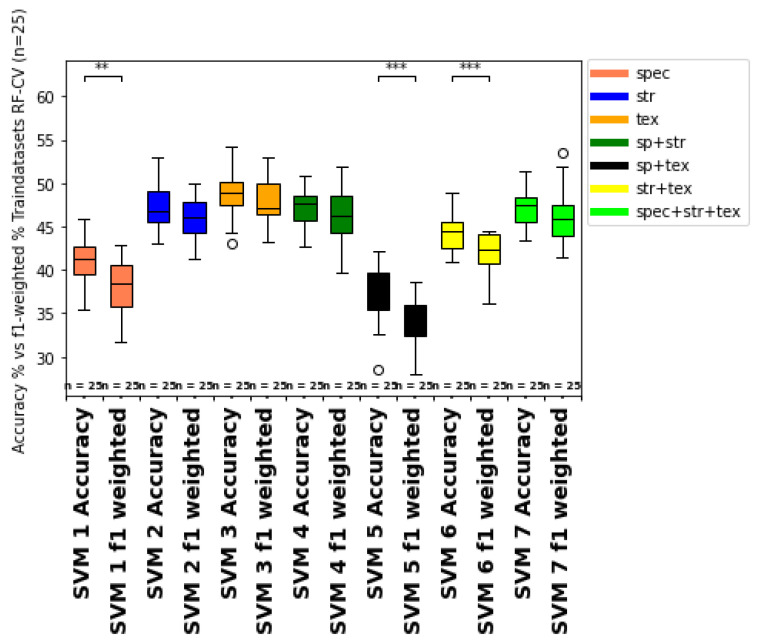
Pairwise statistical comparison of overall accuracy of train data set in % for the SVM classifier with f1-weighted in score in % of the seven different input feature groups (see legend color of grouped boxplots) with significance stars (*) generated according to the Mann–Whitney-U-test between accuracy and f1-weighted, where significant accuracy differences derived from repeated-k-fold cross- validation occurred, with *p*-value significance classes. ** Signal greater than 0.01 (intermediate significance). *** Signal less than 0.001 (vital significance). No stars between boxplots indicate no statistical differences between the model outputs according to Mann-Whitney-U-test.

**Figure 13 sensors-25-00431-f013:**
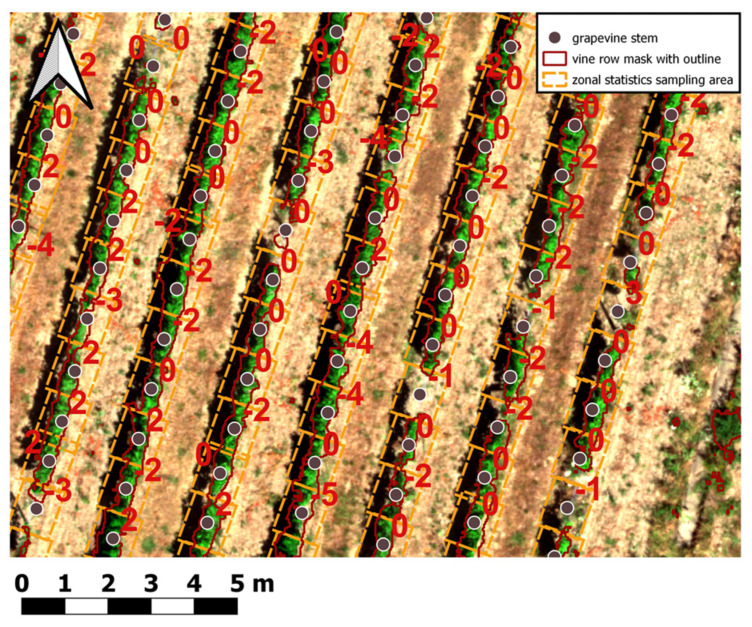
Visualization example of the difference between the ground truth growth classes and the model predicted growth classes from the output from SVM 7 model. Red numbers next to grapevine stems (brown points) with values over zero represent an underestimation of the model prediction compared to ground truth data. In contrast, values less than zero would indicate an overestimation of the growth class model prediction, compared to the ground truth data. Zero values indicate a perfect match of the ground truth with the ML model prediction. The red rectangles represent the area of the zonal statistics aggregation, and the red outline the generated vine row mask (see Section 2.6.9), where the spatial aggregation of the features was achieved. Pixels outside the vine row mask were not considered for spatial aggregation. All mapped geodata are projected to CRS with EPSG: 25,832/ ETRS89/UTM zone 32N.

**Table 1 sensors-25-00431-t001:** Growth and infection classification assessment after Porten [43] with growth classes as integer-coded values and also color-coded for individual growth classes as well as disease/infection type with additive (Additive column) code to describe the infection type; in the description column in parenthesis, the range of the infection strength (INS column) from light (0.5) to severe (4.5) is coded, including the intermediate infection classes 1, 1.5, 2, 2.5, 3, 3.5, and 4. The virus infection class, listed later in some tables, is not listed here due to challenges in visually recognizing virus symptoms with appropriate precision. Within this study, only the growth class of each vine, without the infection description, was used as the target classes for later machine learning model predictions.

Growth Class	Growth Description	Additive	Description	INS
APO/0	Apoplexy/Death/Missing stem	E	Esca symptoms	0.5-4.5
1	Almost dead	S	Blackwood disease (Bois Noir)	0.5–4.5
3	Very weak growth	ES	Esca and Bois Noir	0.5–4.5
5	Medium growth	P	Infection with Peronospora	0.5–4.5
7	Medium to good growth	N	New planted vine/vines	0.5–4.5
9	Excellent-to-excessive growth	NA	Re-expulled from the trunk after APO or other damage	0.5–4.5

Color-Code description: 0—Apoplexy/Death/Missing stem—dark red| 1—Almost dead—red| 3—Very weak growth—orange| 5—Medium growth—cyan, | 7—Medium to good growth—green| 9—Excellent-to-excessive growth)—dark green.

**Table 2 sensors-25-00431-t002:** Spectral properties (spectral band, center wavelength in nanometer, and bandwidth at full width at half maximum (FWHM)) of the different spectral bands of the MicaSenseRed-Edge-MX^TM^. The numbers in brackets after the spectral band name indicate the band number of the generated orthomosaics of this study.

Spectral Band	Center Wavelength (nm)	Bandwidth FWHM (nm)
Blue (1)	475	32
Green (2)	560	27
Red (3)	668	16
Red Edge (4)	717	12
Near Infrared (5)	842	57

**Table 3 sensors-25-00431-t003:** Calculated VI with relevant literature references for vine row extraction (OSAVI), and as input features for the ML-based prediction of the growth classes after [43].

ID	Spectral Indices	Source Reference	Abbreviation
1	Normalized Difference Vegetation Index	[72]	NDVI
2	Normalized Difference Red Edge Index	[73]	NDREI
3	Optimized Soil Adjusted Vegetation Index	[74]	OSAVI
4	Green Normalized Difference Vegetation Index	[75]	GNDVI
5	Transformed Soil Adjusted Vegetation Index	[76]	TSAVI
6	Normalized Difference Water Index	[77]	NDWI

**Table 4 sensors-25-00431-t004:** Utilized structural features used as input features for the ML- based prediction of the growth classes after [43].

ID F.I.	Height and Volume Measures	Name
1	Mean Height	CHM_mean_
2	Median Height	CHM_median_
3	Minimum Height	CHM_min_
4	Maximum Height	CHM_max_
5	Standard Deviation Height	CHM_std_
6	Variance	CHM_var_
7	Aggregated Pixel Volume	CHM Volume

**Table 5 sensors-25-00431-t005:** The Gray-Level Co-occurrence Matrix (GLCM) textural features and their definitions, used as input features for ML-based prediction of the growth classes after [43].

S.N.	Texture Measure	Formula
1.	Mean (ME)	$ME=\sum_{x=0}^{N-1}\sum_{z=0}^{N-1}kP(x,z)$
2.	Variance (VAR)	$VAR=\sum_{x=0}^{N-1}\sum_{z=0}^{N-1}{\left(x-µ\right)}^{2}P(x,z)$
3.	Homogeneity (HOM)	$HOM=\sum_{x=0}^{N-1}\sum_{z=0}^{N-1}\frac{1}{{1+\left(x+z\right)}^{2}}P(x,z)$
4.	Contrast (CON)	$CON=\sum_{x=0}^{N-1}\sum_{z=0}^{N-1}{\left(x-z\right)}^{2}P(x,z)$
5.	Dissimilarity (DIS)	$DIS=\sum_{x=0}^{N-1}\sum_{z=0}^{N-1}P\left(x,z\right)|x-z|$
6.	Entropy (ENT)	$ENT=\sum_{x=0}^{N-1}\sum_{z=0}^{N-1}P\left(x,z\right)log(P\left(x,z\right))$
7.	Angular Second Moment (ASM)	$ASM=\sum_{x=0}^{N-1}\sum_{z=0}^{N-1}{\left(P\left(x,z\right)\right)}^{2}$
8.	Correlation (COR)	$COR=\sum_{x=0}^{N-1}\sum_{z=0}^{N-1}P(x,z)\left[\frac{\left(x-ME)(z-ME\right)}{\sqrt{{VA}_{x}{VA}_{z}}}\right]$

$Note:P\left(x,z\right)=\frac{V\left(x,z\right)}{\sum_{i=0}^{N-1}\sum_{j=0}^{N-1}V};where\;VA\left(x,z\right)\;represents$ the value in the row at the cell x and column z within the moving window. N represents the number of rows or columns in the window.

**Table 6 sensors-25-00431-t006:** Overview table of the input feature groups (1–7) built from the input features of the different feature types (spectral, structural, texture) and their combinations.

ID Input Feature Group	Abbreviation	Input Features
1. spectral features fitted to growth class ground truth	sp	n = 6 NDVI + NDVIRE + OSAVI + GNDVI + NDWI + TSAVI
2. structural features fitted to growth class ground truth	str	n = 4 CHM_(mean)_ + CHM_(max) +_ CHM_(std)_ + CHM_(Volume)_
3. texture features fitted to growth class ground truth	tex	n = 4 Contrast + Correlation + Entropy + Angular Second Moment
4. spectral and structural features fitted to growth class ground truth	sp + str/sp-str	n = 10 NDVI + NDVIRE + OSAVI + TSAVI + GNDVI + NDWI+ CHM_(mean)_ + CHM_(max)_ + CHM_(std)_ + CHM_(Volume)_
5. spectral and texture features fitted to growth class ground truth	sp + tex/sp-tex	n = 10 NDVI + NDVIRE + OSAVI + GNDVI + NDWI + TSAVI + Contrast + Correlation + Entropy+ Angular Second Moment
6. structural and texture features fitted to growth class ground truth	str + tex/str-tex	n = 8 CHM_(mean)_ + CHM_(max)_ + CHM_(std)_ + CHM_(Volume)_ + Contrast + Correlation + Entropy + Angular Second Moment
7. spectral, structural, and texture features fitted to growth class ground truth	sp + str + tex/sp-str-tex	n = 14 NDVI + NDVIRE + OSAVI + GNDVI + NDWI + TSAVI + CHM_(mean)_ + CHM_(max)_ + CHM_(std)_ + CHM_(Volume)_ + Contrast + Correlation + Entropy + Angular Second Moment

**Table 7 sensors-25-00431-t007:** ML classifiers SVM and RFC, with preprocessing steps and selected parameters, parameter settings and parameter ranges for hyperparameter-tuning achieved through grid-search in sklearn, listed for the different classifiers.

Machine Learning Classifiers	Pre-Processing of Input Features	Train–Test Split	Parameter for Hyperparameter Tuning
SVM (Support Vector Machines)	- selection of features - eliminating NoData values - standard scaling of the input features	repeated-k-fold-cross-validation splits = 5, repeats = 5 (n = 25), random state = 1, jobs = −1 with hyperparameter tuning (grid search)	kernel: [linear, rbf]; c: 0.001, 0.01, 0.1, 1, 10, 15, 20, 100, 1000]: gamma: [0.001, 0.01, 0.1, 1, 10]
RFC (Random Forest Classifier)	- selection of features - eliminating NoData values - standard scaling of the input features	repeated-k-fold-cross-validation splits = 5, repeats = 5 (n = 25), random state = 1, jobs = −1 with hyperparameter tuning (grid search)	n_estimators: [25, 50, 100, 150, 300, 500], max_features: [sqrt, log2, none], max_depth: [3, 6, 9, 15, 20, 30], max_leaf nodes: [3, 6, 9] max_samples: [2, 4, 6] min_samples_leaf: [1, 2, 4] criterion: [entropy, gini]

**Table 8 sensors-25-00431-t008:** Results and comparison of model score metrics (accuracy_mean_, f1-weighted mean, accuracy_std_, f1-weighted_std_,) of training and test data fits, received from repeated k-fold cross-validation (n = 25) for the different ML classifiers (SVM, RFC) with grid search-based hyperparameter tuning with repeated-k-fold-cross-validation on the different input feature groups (1–7). For the three best models (SVM4, SVM6, SVM7) according to accuracy of the test data set, the model metrics results are marked with bold green, whereas for the three worst models (RF1, RF3, RF5) according to accuracy of the test data set the metrics are marked in bold red. In the model type column, models trained with only structural features (2) or together with other feature types (4,6,7) are marked in bold gray, to highlight the strong influence of structural features on model classification performance.

	Train Data				Test Data			
Model Type	Accuracy	F1-Weighted	Accuracy std.	F1-Weighted std.	Accuracy	F1-Weighted	Accuracy std.	F1-Weighted std.
**RF 1**	** 31.51% **	** 22.25% **	** 2% **	** 4.40% **	** 30.41% **	** 20.98% **	** 2.79% **	** 4.93% **
**RF 2**	41.58%	32.66%	**3%**	5.29%	40.04%	31.57%	3.33%	5.59%
**RF 3**	** 34.10% **	** 29.10% **	** 2% **	** 5.31% **	** 33.17% **	** 28.32% **	** 2.84% **	** 6.90% **
**RF 4**	37.38%	34.11%	4%	5.51%	37.38%	33.05%	3.99%	6.27%
**RF 5**	** 35.07% **	** 27.20% **	** 3% **	** 3.24% **	** 33.23% **	** 26.79% **	** 4.23% **	** 4.38% **
**RF 6**	39.96%	27.73%	3%	5.14%	38.86%	26.65%	3.65%	4.82%
**RF 7**	38.00%	31.63%	2%	4.53%	36.77%	30.01%	4.37%	6.30%
**SVM 1**	50.48%	46.76%	1%	4.89%	40.65%	37.48%	3.16%	3.40%
**SVM 2**	50.34%	51.23%	2%	2.41%	47.38%	46.21%	1.89%	2.76%
**SVM 3**	40.13%	52.49%	2%	2.64%	37.98%	47.72%	2.56%	2.90%
**SVM 4**	** 51.75% **	** 49.63% **	** 2% **	** 1.63% **	** 47.42% **	** 45.86% **	** 2.78% **	** 3.12% **
**SVM 5**	50.13%	35.68%	3%	1.67%	44.21%	33.59%	1.92%	2.44%
**SVM 6**	** 51.05% **	** 47.38% **	** 2% **	** 3.47% **	** 47.57% **	** 42.39% **	** 2.69% **	** 2.97% **
**SVM 7**	** 53.66% **	** 50.11% **	** 2% **	** 2.27% **	** 48.51% **	** 45.50% **	** 2.22% **	** 3.16% **

**Table 9 sensors-25-00431-t009:** Confusion matrix of the test data set for the best overall accuracy prediction score (SVM 7) with 48.51% test accuracy and 53.7% train accuracy from repeated- k-fold cross-validation with tuned hyperparameter. The color marked cells represent the colors according to the color-coding after [43], including neighbor cells of each growth class (to improve the contrast of the values in these cells these are marked in bold to show the percentage of overlapping neighbor growth classes). Letter P in the horizontal bar of the growth classes (GC) stands for the predicted value view (GP), whereas letter L in the vertical bar stands for the labeled growth class view (GL) for each growth class (0-9). Additionally, the UA (User Accuracy) and PA (Producer Accuracy) were calculated for interpretation of classification results and model evaluation (see following text in discussion). The column, and row headers as well as the values in the PA row and the UA column are marked in bold to receive better contrast for reading, and to show their importance for interpreting the class specific accuracies.

GC	0 GP	1 GP	3 GP	5 GP	7 GP	9 GP	UA
**0 GL**	**29.28**	**34.25**	**27.62**	6.63	2.21	0.00	**29.28**
**1 GL**	**14.77**	**36.29**	**29.11**	15.61	4.22	0.00	**36.29**
**3 GL**	3.69	**10.82**	**35.88**	**44.59**	5.01	0.00	**35.88**
**5 GL**	0.00	1.08	**16.81**	**65.73**	**16.16**	0.22	**65.73**
**7 GL**	0.30	0.30	2.72	**39.58**	**57.10**	**0.00**	**57.10**
**9 GL**	0.00	0.00	0.00	9.09	**81.82**	**9.09**	**9.09**
**PA**	**60.95**	**43.86**	**32.00**	**36.27**	**34.29**	**18.92**	**SVM 7**

## Data Availability

The data presented in the study will be available from the corresponding author upon request.

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
