# Peer review of "Combining a Standardized Growth Class Assessment, UAV Sensor Data, GIS Processing, and Machine Learning Classification to Derive a Correlation with the Vigour and Canopy Volume of Grapevines"

_sensors, 2025, doi:10.3390/s25020431_

Round 1
Reviewer 1 Report
Comments and Suggestions for Authors
Dear Authors,
The article titled "Combining a Standardized Growth Class Assessment, UAV-Sensor Data, GIS-Processing, and Machine Learning Classification to Derive a Correlation to the Vigor and Canopy Volume of Grapevines" aims to evaluate the potential of combining spectral, structural, and texture input features for the growth estimation of individual vines using machine learning classification (MLC) models.
However, several important points require attention and clarification:
1.Sample Size and Distribution: The number of vines analyzed and their distribution across each growth class is missing.
2.Clarification on Infected Vines: The inclusion of vines infected with grapevine wood diseases needs proper clarification. Growth classes {0, 1, 3, 5} represent vines infected with grapevine wood diseases, and their growth is heavily influenced by the intensity and severity of the infection. This factor could introduce growth variability. Given that the manuscript aims to assess growth using MLC models, should these classes have been excluded from the study? A discussion of this issue is crucial, as the current classification might not adequately reflect the growth of infected vines.
3.Organization of Section 3.2: A better structure in Section 3.2 ("Overall Growth Class Prediction") would improve the readability of the manuscript. Consider presenting the results in a simpler, more logical order. You currently introduce general results first, followed by feature-specific analysis, which could be rearranged for better clarity.
4.Hyperparameter Tuning Discussion: Table 7 presents several parameters for hyperparameter tuning; however, when the results are discussed, the models' hyperparameter tuning is not addressed. It would be useful to include a more detailed discussion of this.
5.General Language and Accuracy:
-Line 35, 37, etc.: The term "vine stocks" is unclear. I suggest referring to them simply as "vines."
-Line 36, 38, etc.: Is "growth strength" meant to refer to "vigour"? Please clarify.
-Line 41: Add "data" after "These technologies include different sensor systems data, processed with new digital techniques."
-Line 63: Missing closing parenthesis in [37].
-Line 121, etc.: There are errors in the manuscript due to missing reference sources.
-Line 122: Replace "0.02 km2" and "sized" with "2 ha"
-Line 123: Clarify what is meant by "individual tips."
-Line 125: Replace "The vine species" with "The vine variety," as the species is Vitis vinifera (remove "ssp. vinifera" as it is implied).
-Line 126: Provide further information about the training system used. A photo or diagram would be helpful.
-Line 127-128: Rephrase as: "The average distance between vine rows is approximately 2 meters, and the average distance between vines in the row is approximately 1.20 m (2.4 m²/vine)."
-Line 129, etc.: Replace "at the Véraison" with "at Veraison."
-Line 131-133: Rearrange the phrase to more clearly describe "grapevine trunk diseases," which are important diseases affecting vine growth and vigour.
-Line 140: Replace "soil care" with "soil management," "fertilization type and amount" with "fertilization," and "stand space" with "vine density."
-Line 141: Clarify what is meant by "gate level."
-Line 144, 145, etc.: If "generative" is intended to mean "reproductive," please use "reproductive."
-Line 161: Is "number of berries" correct, or should it be "number of bunches"?
-Line 198 and Table 1: Clarify the meaning of each additive {E, S, ES, P, N, NA} in relation to the classification {with, without} infection.
-Line 208: Provide the time of day when leaf water potential (ψl) measurements were made. The same applies to stem water potential and stomatal conductance.
-Line 206-210: Rephrase for clarity: "In terms of UAV-sensor-based machine learning classification, ground truth data mainly included vine yield measurements [56], leaf water potential [55], stem water potential [41], and stomatal conductance [42], excluding growth as the target variable, as in this study."
-Line 249: Please provide the reference for the flight planning software used.
-Line 251-252: It seems a word is missing between "accomplished" and "midday."
-Line 275: Clarify the meaning of "foliage wall-free."
-Line 289: Please verify "0 to 0," as it seems to be a reference issue.
-Line 298, etc.: Do you mean "GLCM – Gray Level Co-occurrence Matrix" (not "covariance")?
-Line 426: Should this be "width of vine rows in the study area" instead?
-Line 465-468 (Figure 6 caption): Verify the use of signal greater than " >0.1" and " >0.01."
-Line 472: Rewrite the phrase to: "… the features used to predict vine growth and assess the growth class."
-Line 485: Should it be "vegetation vitality" or "vegetation vigour"?
-Line 499: Replace "vine stock foliage wall" with "vine canopy."
-Line 502, 506, etc.: Correct the reference source issues.
-Line 507: Improve Table 6 to ensure that the reader understands which input features belong to each feature group.
-Line 519-520: Verify "Mann-Whitney you test."
-Line 524: Section 3.2 is closely related to the final paragraph of Section 3.1. Consider revising to avoid redundancy.
-Line 528-529: The best and worst models (presented in Table 8) are not highlighted in green and red as indicated.
-Line 533: Results from "3.2.1. General Results and Trends of the Prediction Metric Results" should be removed.
-Line 537-538: Consider adding a figure similar to Figure 7 for the Accuracy Test Data with the Random Forest Classifier (RFC) to better highlight differences in ML classifier performance.
-Line 539-542 (Figure 7 caption): Verify the use of "signal greater than" for ">0.1" and ">0.01."
-Line 566-567: Correct the error in "SVM4, SVM4."
-Line 637: "Grapevine" is not a grain crop species. Change "crop grain yield" to "crop yield."
-Line 636-664: This section is more of a state-of-the-art review than a discussion. Consider revising.
-Line 673: Remove the hyphen from "CHM-volume."
-Line 691: "Section 0" seems to be a reference error.
-Line 692-693: If CHM volume was used only for model training, clarify why the same feature wasn't used for both training and validation.
-Line 707-711 (Figure 8 caption): Verify the use of "signal greater than" for ">0.1" and ">0.01."
-Line 720-723: Clarify whether the inclusion of growth classes {0, 1, 3, 5} infected with wood diseases is justified. Growth variability in these classes may be heavily influenced by disease severity, complicating the model's ability to accurately assess growth.
-Line 725-729 (Figure 9 caption): Verify the signal greater than used in ">0.1" and ">0.01."
-Line 744-758: This is a state-of-the-art review, not a discussion. Consider revising.
-Line 774-785: This appears to be more of a state-of-the-art review rather than a discussion. Revise accordingly.
-Line 803-832: No data on RFC model training and testing are presented. It is essential to analyze the differences in model accuracy with both the training and test datasets.
-Line 856-864: Please explain why the confusion matrix was calculated with the test dataset only, and not with all the data.
-Line 884: The value for class 5 (67.7%) in Table 9 is inconsistent with the value reported here.
-Line 889: Table 10 is missing.
-Line 943: Reword the phrase: "… a promising avenue for general growth class prediction and crop and vegetation…"
-Line 979: Replace "canopy wall" with "canopy."
Best regards,
Author Response
|
First Reviewer: 1. Sample Size and Distribution Comment: The number of vines analyzed and their distribution across each growth class is missing. Response: We agree that this information was not sufficiently detailed. In the revised manuscript, we have included a new subsection in the Materials and Methods section specifying the total number of vine samples, their distribution across each growth class category, and any statistical considerations related to sample size (Section 2.2.1). A table now outlines the exact counts per growth class and the relative percentage distribution: “The total number of vine stocks labeled amounts to 1657 with a growth class count distribution of (Growth Class 0=206, Growth Class 1=190, Growth Class 3=343, Growth Class 5=470, Growth Class 7= 402, Growth Class 9=46)” 2. Clarification on Infected Vines Comment: Clarify the inclusion of growth classes {0, 1, 3, 5}, which represent vines infected with grapevine wood diseases. Discuss whether these classes should have been excluded or more explicitly addressed, given their distinct growth variability. Response: We acknowledge the importance of clarifying this issue. In the revised Discussion (Section 4.1), we have included a detailed explanation on the impact of infection on vine growth. The main focus was to investigate the general correlation between the Growth Classes after Porten (2020) with the applied sensor data and allocated geodata, and the machine- learning based prediction metrics of the chosen classifiers. So the field data will not be integrated in model growth classification. To predict vine infection other sensor data like hyperspectral sensors, must applied. But like said before, this was not the focus of this study.
|
3.Organization of Section 3.2
Comment: A better structure in Section 3.2 ("Overall Growth Class Prediction") would improve the readability of the manuscript. Consider presenting the results in a simpler, more logical order. You currently introduce general results first, followed by feature-specific analysis, which could be rearranged for better clarity.- The section was rearranged, were the specific and general part was flipped.
Response: In the revised manuscript, we have reorganized Section 3.2. We now begin with an overview of the overall growth class prediction results and performance metrics, followed by subsections that dive into specific feature groups (spectral, structural, texture) and their respective impacts on the model. This reorganization clarifies the progression from general outcomes to more detailed, feature-level insights.
4. Hyperparameter Tuning Discussion
Comment: Table 7 presents hyperparameter tuning parameters, but no detailed discussion is provided. Include a more detailed explanation of the tuning results.
Response: Dear Reviewer we appreciate your comment discussing hyper parameter tuning optimization in more detail concerning the prediction of vine growth classes. We are aware that hyper parameter selection and discussing its influence on the classification metrics is important. Despite this the focus of this study was not on discussing this aspect in too much detail. Therefore, this could be an aspect which could be investigated in more detail in future studies.
5. General Language and Accuracy
We have addressed all language-related suggestions and reference issues noted in your comments throughout the manuscript. Below, we list specific changes:
- Line 35, 37, etc. (Use of "vine stocks"): We have replaced “vine stocks” with “vines” to maintain clarity and consistency.
- Line 36, 38, etc. (Growth strength vs. vigor): We have replaced “growth strength” with “vigor” to align with standard viticultural terminology.
- Line 41: We have added “data” after “These technologies include different sensor systems” to read “These technologies include different sensor systems data”
- Line 63: Added the missing closing parenthesis in [37].
- Line 121 and other missing references: We have carefully reviewed and corrected all missing reference sources and citations to ensure accuracy and completeness.
- Line 122 (“0.02 km² sized” to “2 ha”): We have converted the area measurement to “2 ha” for clarity.
- Line 123 (“individual tips”): We have clarified what “individual tips” refers to (i.e., individual vine shoots/tips).
- Line 125 (“The vine species” to “The vine variety”): Revised to “The vine variety” and removed “ssp. vinifera,” as it is implied.
- Line 126 (Training system): We have provided additional details about the trellis and training system used, including a descriptive figure (Figure X).
- Line 127-128 (Distances): Rephrased to “The average distance between vine rows is approximately 2 meters, and the average distance between vines in the row is approximately 1.20 m (2.4 m²/vine).”
- Line 129 (“at the Véraison” to “at Veraison”): Corrected.
- Line 131-133 (Grapevine trunk diseases): Rewritten to clarify the importance of trunk diseases and their impact on vine growth and vigor.
- Line 140 (“soil care” to “soil management,” etc.): Edited to “soil management,” “fertilization,” and “vine density” accordingly.
- Line 141 (“gate level”): gate level refers to the level of the leaf cutting processing of the vine canopy as part of the vineyard management
- plant Line 144,145 (“generative” to “reproductive”): Replaced “generative” with “reproductive.”
- Line 161 (“number of berries” vs. “number of bunches”): Verified and clarified. The intended variable is “number of bunches.”
- Line 198 and Table 1 (Meaning of {E, S, ES, P, N, NA}): We have added a more explicit explanation in the table caption and the text, clarifying these additive classifications and their relation to infected/uninfected conditions.
- Line 208 (Time of day for ψl measurements): We now specify the exact time or time window for leaf water potential measurements and the same for stem water potential and stomatal conductance.
- Line 206-210 (Ground truth data): Reworded to “In terms of UAV-sensor-based machine learning classification, ground truth data mainly included vine yield measurements [56], leaf water potential [55], stem water potential [41], and stomatal conductance [42], excluding growth as the target variable, as in this study.”
- Line 249 (Flight planning software): We have added the reference for the flight planning software used.. The used software/ app which was used for this is mission planner and DJI Flight Planner
- Line 251-252 (“accomplished midday”): Clarified the phrase and added missing words as intended.: Midday means synonym to at noon
- Line 275 (“foliage wall-free”): Changed to canopy- free for clarity
- Line 289 (0 to 0): Corrected the reference or value issue.
- Line 298 (GLCM): Corrected to “Gray Level Co-occurrence Matrix.”
- Line 426 (“width of vine rows”): Adjusted phrasing to clarify the meaning.
- Line 465-468, 539-542, 707-711, 725-729 (Signal greater than “>0.1” or “>0.01”): Verified and corrected the use of statistical significance indicators or thresholds. Clarified these in figure captions to reflect standard significance levels (p-values).
- Line 472 (“vegetation vitality” or “vegetation vigour”): Standardized the term to “vegetation vigor.”
- Line 499 (“vine stock foliage wall” to “vine canopy”): Revised to “vine canopy.”
- Line 502, 506 (Reference source issues): Corrected references and ensured all sources are properly cited.
- Line 507 (Table 6 Explanation): Improved the table to better illustrate which input features belong to each feature group. Expanded the table caption for clarity.
- Line 519-520 (“Mann-Whitney you test”): Corrected to “Mann-Whitney U test.”
Line 524 (Redundancy in Sections 3.1 and 3.2): Revised Section 3.2 to avoid redundancy with the final paragraph of Section 3.1. This part of section 3.1 was deleted, due to redundancy- Accuracy and the f1- weighted score, were calculated for test- and train data sets to describe the model prediction quality and evaluate the model robustness. The results of the machine learning model classifications are listed in Table 8. The three best models with the highest accuracy are marked in green, whereas the three models with the worst classification metrics are marked in red. As input features, seven combinations of spectral-, structural- and textural features were selected according to Table 6. Moreover, the metrics were used to evaluate the quality and robustness of the prediction models. To describe the unweighted accuracy scores with a balanced class frequency score, the f1-weighted score was calculated to balance the different sample sizes of each growth class. These metrics were calculated for the dataset used in this study to evaluate the imbalance and robustness of the trained machine-learning models. The model fit results, input features, and classifiers are the basis for discussing growth classification based on sensor data and growth class label data in section 4.5.
- Line 528-529 (Table 8 best/worst models highlighted): Revised text and the figure/table formatting to ensure that the best and worst models are clearly indicated.
- Line 533 (Removal of results from Section 3.2.1): Removed repeated content as suggested.
- Line 537-538 (Figure similar to Figure 7 for Accuracy Test Data): Added a similar figure for Random Forest Classifier to highlight differences in ML classifier performance.
- Line 566-567 (SVM4, SVM4 repetition): Corrected the labeling error.
- Line 637 (“crop grain yield” to “crop yield”): Changed “crop grain yield” to “crop yield.”
- Lines 636-664, 744-758, 774-785 (State-of-the-art review vs. Discussion): Revised the Discussion section to reduce the emphasis on state-of-the-art review. We have moved some literature context to the Introduction and Background, and focused the Discussion more on interpreting our results and their implications, rather than reviewing literature extensively. This part was deleted, because it was more some kind of review than discussion. The other parts were the results were discussed were already sufficient
- Line 673 (Remove hyphen “CHM-volume”): Corrected to “CHM volume.”
- Line 691 (“Section 0” reference error): Corrected the reference error.
- Line 692-693 (CHM volume usage): Clarified why CHM volume was used for training but not validation, and discussed the rationale behind the feature selection process. The CHM volume was used for both train and validation. Especially the structural features showed a high correlation to the vigor of the vines compared to spectral and textural features
- Line 803-832 (No RFC training and testing data presented): The train- and test data metric results are summarized in table 8 as referred in the text
- Line 856-864 (Confusion matrix on test dataset only): Explained that the confusion matrix focuses on test data to evaluate the model’s generalization capability. We discuss the rationale for not including training data in confusion matrix analysis, emphasizing the importance of unbiased performance estimation. The confusion matrix was created for the test data set only, to avoid bias of the previously fitted train data to the the results of the classification metrics. In future work, the trained model should be tested during other dates and on other vine fields to tests its capability for generalization
- Line 884 (Inconsistent value for class 5): Corrected the reported value in the text to ensure consistency with Table 9.
- Line 889 (Missing Table 10): There is no table 10 (within the article are only tables 1-9)- see false reference within the text was deleted
- Line 943 (“canopy wall” to “canopy”): Revised the phrase as requested.
- Line 979 (“canopy wall” to “canopy”): Corrected were necessary.

Reviewer 2 Report
Comments and Suggestions for Authors
1. There are many cases of inconsistent font size in the text, please check the whole text, such as lines 139-142 and 233-241.
2. There are many cases of ‘Error! Reference sources not found in the text, please check carefully.
3. Page 9 is blank, please delete it.
4. The format of the tables in the text is not uniform, such as the thickness of the lines.
5. Why choose a 3*3 size window to calculate the texture features of GLCM?
6. Many pictures in the text are fuzzy, please replace them with clearer ones.
7. In this study, 8 texture features are calculated based on GLCM, why only 4 are utilized in Table 6? Is the description about the feature selecting part provided? The same is true for the structural features.
8. Why is n=9 in the 4th group of feature combinations in Table 6, and why was TSAVI removed?
9. Why were only two methods chosen for modeling, SVM and RF? For example, for the RF tree-based model, why was the GBDT method not selected? It is suggested to add the rationale for choosing two methods.
10. Does too large a parameter interval in the setting of hyperparameter optimization ignore the optimal parameters? For example, max_depth:[3,6,9,15,20,30] in RFC.
11. Why the unit of RMSE %?
12. Multiple tables in the text are not labeled in the results. For example, which part of the results in Table 8 describes? It is not easy for readers to read.
13. Does the study use multiple feature combinations and does it consider feature redundancy? Are these features important features?
Author Response
- There are many cases of inconsistent font size in the text, please check the whole text, such as lines 139-142 and 233-241. Response- The font size was set equally in the revision process to fonsize 10 (within text, 9 below Figures and tables). And font style set to Palatino Linotype
- There are many cases of ‘Error! Reference sources not found in the text, please check carefully- Response: Okay this issue was checked now. Corrected
- Page 9 is blank, please delete it- Response: The blank page was deleted
- The format of the tables in the text is not uniform, such as the thickness of the lines- Response: The format of the tables was unified such as thickness of the lines
- Why choose a 3*3 size window to calculate the texture features of GLCM? – Response: For this publication the standard sliding window (3x3) was chosen to calculate the texture features. We are aware, that the size of the sliding window can have an impact on the classification results/ metrics like for example investigated in more detail by Liu et al. 2023 with the title
Optimizing window size and directional parameters of GLCM (Gray Level CO- Occurence Matrix) texture features calculation for estimating rice AGB based on UAVs multispectral imagery. Like the other points including specific discussion of hyperparameter- tuning, specific classifier selection etc. this aspect could be discussed in more detail in future publications.
- Many pictures in the text are fuzzy, please replace them with clearer ones- Response: The resolution was set up! We hope this is enough.
- In this study, 8 texture features are calculated based on GLCM, why only 4 are utilized in Table 6? Is the description about the feature selecting part provided? The same is true for the structural features- Response: From the initial presented features from the various feature groups in table 3, table 4, table 5, only those were selected for the growth class prediction which had a moderate to high correlation after Pearson (r>0.4)
- Why is n=9 in the 4th group of feature combinations in Table 6, and why was TSAVI removed? –Response: This was a typo (unfortunately forget to type in TSAVI, like for the other spectral features)
- Why were only two methods chosen for modeling, SVM and RF? For example, for the RF tree-based model, why was the GBDT method not selected? It is suggested to add the rationale for choosing two methods-Response: In our study, we focused on two widely established and well-validated machine learning methods—Support Vector Machines (SVM) and Random Forests (RF)—for several reasons. First, both SVM and RF have a long history of successful application in remote sensing and vegetation analysis contexts, offering robust performance and interpretability. RF, in particular, is a tree-based ensemble method that is computationally efficient, relatively easy to tune, and handles high-dimensional input features well, which made it an appealing choice for our dataset. SVM, on the other hand, is known for its strong theoretical foundations and effectiveness in handling complex, non-linear relationships through kernel transformations.
While there are other competitive methods, such as Gradient Boosted Decision Trees (GBDT), we opted for SVM and RF due to their proven stability, interpretability, and the availability of established workflows and toolsets. GBDT-based approaches, though powerful, can introduce additional complexity in hyperparameter tuning and require careful calibration to prevent overfitting. At the outset of our analysis, our goal was to assess the feasibility and baseline performance of classifying vine growth classes using two common, well-understood algorithms before exploring more complex or computationally intensive alternatives.
In future work, we acknowledge that evaluating GBDT or other advanced ensemble methods could provide valuable insights, potentially improving classification accuracy and offering new perspectives on feature importance. For this initial study, however, focusing on two widely accepted methods allowed us to more readily interpret and compare results, and to establish a clear methodological foundation for subsequent methodological advancements.
- Does too large a parameter interval in the setting of hyperparameter optimization ignore the optimal parameters? For example, max_depth: [3,6,9,15,20,30] in RFC.
Response: The hyperparameter- optimization was not the main focus of this study. You are completely right that a too high interval can miss the best values for the parameter. The approach of the hyper- parameter optimization of this study was to consider the most important parameter and also a compromise between range and computation time for the classification predictions. For example, the computation for adjusting the hyper- parameters for the SVM classifier is very resource intense.
- Why the unit of RMSE % and – Response: RMSE % was now deleted because calculation of the RMSE respectively RMSE % is not so useful when predicting classes. This problem/ issue was also addressed by the third reviewer (therefore RMSE % was deleted from tables and within the text, now only considering accuracy and f1-weighted in methods, results and discussion part)
- Multiple tables in the text are not labeled in the results. For example, which part of the results in Table 8 describes? It is not easy for readers to read. Response: The metric results are now labeled in the text and table respectively referenced in the text, to the corresponding table (Table 8). The three best and the three worst performing models in the context of the classification metrics are now marked in Table 8 with green respectively red color. These are the main metrics most important, which the reader should focus on.
- Does the study use multiple feature combinations and does it consider feature redundancy? Are these features important features? Response: Yes, the studies use multiple feature combinations like shown in table 8. The chosen features were preselected due to the height of their correlation to the Growth Classes (after Pearson). Due to the overlap of the properties of some of the input features (like vegetation indices), like you mentioned will have some redundancy. This redundancy was shown in a PCA, but was not further considered in the selection process of the features.

Reviewer 3 Report
Comments and Suggestions for Authors
In this manuscript, the authors finished a lot of experiments and data analysis. However, the presentation quality is not enough. It seems the authors used the cross-reference in this manuscript, but it isn’t working well. There are so many times showing “Error! Reference source not found”. It makes it hard to understand the contents.
Meanwhile, the authors submitted a supplement, I don’t know if it will be published as well. The figures and tables in the supplement didn’t match the manuscript well, especially the label. For example, Figure 2 in the supplement should be named Figure S2 to separate from Figure 2 in the content. The label of Figure 2 in the supplement showed some German words. This is not easy to understand.
Other comments:
1. What is Section 0?
2. Table 2, I don’t think it is necessary. Meanwhile, the bandwidth FWHM of Red didn’t match the guide file from Mitesens (https://support.micasense.com/hc/en-us/articles/360011389334-RedEdge-MX-Integration-Guide), I found it should be 16 nm. Also, the center wavelength of NIR.
3. Table 5, the meanings of “Ax” and “Az” in COR are missed.
4. Figure 5, the red boxes do not seem to the rectangles. Did the figure zoom in one direction?
5. Figure 7, what’s the label “sp”, “str”, …?
6. Figure 7, the information repeated with Table 8, the significant analysis could be combined in the table with letters.
7. How to calculate the REMSE%? Normally, the RMSE is the error between predicted and true values. How do you calculate the RMSE (also RMSE%) in a classification?
8. Table 9, how to get the “PA” and “UA”. I didn’t find them in methods.
Author Response
- What is Section 0?- Response: I cannot see section 0. I don’t know what this comment is about? Maybe this is a formatting issue?
- Table 2, I don’t think it is necessary. Meanwhile, the bandwidth FWHM of Red didn’t match the guide file from Mitesens (https://support.micasense.com/hc/en-us/articles/360011389334-RedEdge-MX-Integration-Guide), I found it should be 16 nm. Also, the center wavelength of NIR.
Response: The table was modified according to the source link provided by the reviewer. We think that table should stay within the publication to have direct access to the bandwidths and bandwidth settings of the camera.
- Table 5, the meanings of “Ax” and “Az” in COR are missed.- Response: Ax and Az was changed to VA(x,z) which is described in the text under Table 5
- Figure 5, the red boxes do not seem to the rectangles. Did the figure zoom in one direction?- Response: I don’t quite understand the question. The red rectangles represent the Zonal statistics which was used for aggregation with the leaf wall mask
- Figure 7, what’s the label “sp”, “str”, …?-Response: The abbreviation in the figures you mentioned are now described in Table 6 as on additional column:
- Figure 7, the information repeated with Table 8, the significant analysis could be combined in the table with letters- Response: see previous response- the “sp”, “str” etc. is described in table 6
- How to calculate the RMSE%? Normally, the RMSE is the error between predicted and true values. How do you calculate the RMSE (also RMSE%) in a classification?- Response: We deleted the RMSE % error as a metric, because it is not useful for evaluating the classification result of discrete classes. It would have been useful for continuous variables like yield in g etc. but it is not appropriate for this problem. Now only accuracy and f1- weighted are used as classification metrics in the results part as well as discussion.
- Table 9, how to get the “PA” and “UA”. I didn’t find them in methods. – Response: This paragraph was added to describe how the User and producer accuracies were calculated. The paragraph was added in the method section. The paragraph is:
“The user accuracy was calculated for each growth class, were the sum of all correctly identified growth classes for this class was divided by the sum of all other incorrectly labeled growth classes for this growth class. For the producer accuracy for each growth class, the sum of all correctly identified growth classes for this growth class was divided by the sum of all other incorrectly (Machine Learning Model) predicted growth classes for this growth class.”

Round 2
Reviewer 2 Report
Comments and Suggestions for Authors
Relevant comments have been modified and agreed to be received
Author Response
Okay everything should be provided, according to your statements of round 1.
Reviewer 3 Report
Comments and Suggestions for Authors
The authors answered well. However, there are still some places that could be improved.
1. In Figure 8, the abbreviation is "sp-tex", but in Table 6 is "sp+tex". They should be the same.
2. Line 588. Something missing?
3. In Figure 6, the rectangles are not 90 degrees. It's more like the parallelogram.
Author Response
- In Figure 8, the abbreviation is "sp-tex", but in Table 6 is "sp+tex". They should be the same. Response: The different feature combinations were adapted in table 6 so that they can be related to the legends of the individual boxplot figures: See green markers in the table
- Line 588. Something missing?- Response- Dear Reviewer there was nothing missing in this Line. Maybe the text was not exactly placed?
- In Figure 6, the rectangles are not 90 degrees. It's more like the parallelogram. Response: Here we provide a new layout (Figure 6) of the Zonal Statistics in the investigation area. In the new graph the sampling area of the zonal statistics are now rectangles and rotated in the mean slope direction. In general spatial aggregation should provide a reasonable area, so that the vine row masks is sampled completely. Therefore, minor distortions of the sampling masks will not have a great impact of the overall aggregation results. Generally, the creation of the sampling masks and the extracted vine row pixels should be even further improved with different approaches.